# AION-1: Omnimodal Foundation Model for Astronomical Sciences

**Liam Parker**[*,1,2,3,4], **Francois Lanusse**[*,5,2], **Jeff Shen**[*,6], **Ollie Liu**[7], **Tom Hehir**[8], **Leopoldo Sarra**[2], **Lucas Meyer**[2], **Micah Bowles**[9], **Sebastian Wagner-Carena**[2,3], **Helen Qu**[2], **Siavash Golkar**[2,3], **Alberto Bietti**[2], **Hatim Bourfoune**[10], **Pierre Cornette**[10], **Keiya Hirashima**[2,11], **Geraud Krawezik**[2], **Ruben Ohana**[2], **Nicholas Lourie**[3], **Michael McCabe**[2,3], **Rudy Morel**[2], **Payel Mukhopadhyay**[1,8], **Mariel Pettee**[12], **Bruno Regaldo-Saint Blancard**[2], **Kyunghyun Cho**[3], **Miles Cranmer**[8], **Shirley Ho**[2,3,6]

**The Polymathic AI Collaboration**

[1]University of California, Berkeley, [2]Flatiron Institute, [3]New York University, [4]Lawrence Berkeley National Laboratory, [5]Université Paris-Saclay, Université Paris Cité, CEA, CNRS, AIM, [6]Princeton University, [7]University of Southern California, [8]University of Cambridge, [9], Univeresity of Oxford, [10]IDRIS, CNRS, [11]RIKEN Center for iTHEMS, [12]University of Wisconsin, Madison

## Abstract

While foundation models have shown promise across a variety of fields, astronomy lacks a unified framework for joint modeling across its highly diverse data modalities. In this paper, we present AION-1, the first family of large-scale multimodal foundation models for astronomy. AION-1 integrates a large number of heterogeneous data types using a two-stage architecture: modality-specific tokenization followed by transformer-based masked modeling of cross-modal token sequences. Trained on over 200M astronomical objects, AION-1 demonstrates strong performance across regression, classification, generation, and object retrieval tasks. Beyond astronomy, AION-1 provides a scalable blueprint for multimodal scientific foundation models that can seamlessly integrate heterogeneous combinations of real-world observations. Our model release is entirely open source, including the dataset, training script, and weights: `https://github.com/PolymathicAI/AION`.

## 1 Introduction

Foundation models have transformed natural language processing and computer vision [1, 20, 18]. However, they have not been fully explored in scientific domains where data are often complex and heterogeneous, combining multiple instruments, measurement protocols, and noise sources unique to real-world experiments. As a result, many scientific analyses employ bespoke models that treat each modality in isolation or rely on strict - often hand-crafted - schemas for cross-modal data fusion.

Within the broader scientific landscape, astronomy provides a particularly compelling testbed for the development of multimodal scientific foundation models owing to both the volume of publicly available data and its extraordinary diversity of measurements. Indeed, recent works have begun to explore multi-modal foundation models in astronomy [48, 43, 80, 50]; however, these approaches have been limited to single physical phenomena and relied primarily on contrastive objectives, which face fundamental limitations including generalization to arbitrary modalities and difficulty in capturing information beyond the mutual information between modalities.

---

[*]Denotes equal contribution. Contact: `lhparker@berkeley.edu`, `francois.lanusse@cnrs.fr`

39th Conference on Neural Information Processing Systems (NeurIPS 2025).

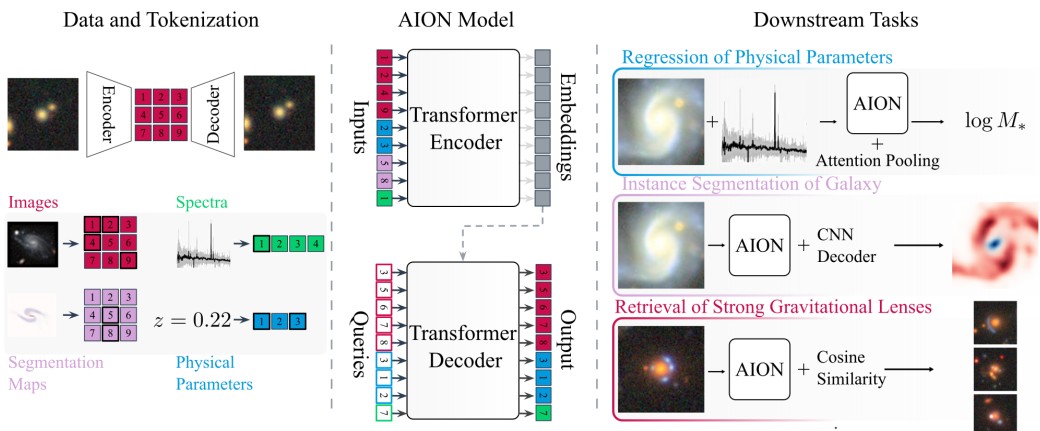

Figure 1: AION-1 integrates **39 different data modalities** — multiband images, optical spectra, and various properties and measurements — into a single model usable for a wide range of downstream applications. It implements a two-step process: first, **bespoke tokenization strategies** that homogenize the diverse scientific data, followed by **multimodal masked modeling** that learns how different observations relate, inducing a deep understanding of the underlying physical objects. Astronomers can then leverage AION-1's rich astrophysical understanding for a variety of downstream tasks.

In this paper, we introduce AION-1 (AstronomIcal Omni-modal Network), the first large-scale multimodal foundation model for astronomy designed to handle *arbitrary* numbers of modalities across multiple physical phenomena. AION-1 unifies imaging, spectroscopy, photometry, and other object-level measurements from major ground- and space-based observatories into a single model for galaxies, stars, and quasars. By bridging these disparate data types, AION-1 addresses a key challenge in scientific machine learning: the integration of multiple heterogeneous datasets spanning different instruments, measurement protocols, noise sources, and physical phenomena into a single, unified framework.

At the heart of AION-1 lies a two-step approach: **Universal Tokenization of Diverse Data**, where we homogenize real-world scientific observations with discrete quantization across different data types, instruments, and observatories, followed by **Multimodal Masked Modeling**, where we train a single transformer encoder-decoder with a masked-token objective over all modalities simultaneously. Once trained, we demonstrate emergent behaviors in the AION-1 models that reflect the potential for multimodal scientific foundation models to capture non-trivial physical insights from raw data alone:

- **Emergent Physical Understanding.** AION-1 can solve non-trivial scientific tasks using only a simple linear head on top of its learned representations.

- **Superior Performance in the Low-Data Regime.** AION-1 achieves competitive results on downstream tasks even with orders of magnitude less data than its supervised counterparts.

- **Flexible Data Fusion.** AION-1 can use arbitrary combinations of observations, enabling seamless data fusion on downstream tasks as well as cross-modal conditional generation.

- **Physical Structure of the Latent Space**: AION-1's embedding space organizes objects along physically meaningful directions, enabling powerful retrieval of rare observations that surpasses current state-of-the-art retrieval methods in astronomy.

Beyond astronomy, the data tokenization strategies, masked modeling, and cross-modal generation strategies introduced address key challenges in real-world scientific data: namely, heterogeneity, noise, and instrument-specific idiosyncrasies. Moreover, by focusing on purely observational data, our approach is applicable in any data-rich field, even when strong physical models are not available.

## 1.1 Contributions

In summary, we present the following contributions:

- We introduce AION-1, a family of token-based multimodal scientific foundation models ranging in size from 300M to 3.1B parameters. AION-1 is designed for arbitrary combinations of highly heterogeneous scientific observations.

- We develop bespoke tokenization methods to homogenize a wide variety of astronomical data into a single coherent corpus. These innovations address the heterogeneity, noise, and instrument-specific peculiarities that challenge standard scientific modeling.

- We demonstrate that AION-1 achieves competitive to state-of-the-art performance on a broad range of scientific tasks with even simple probing, while significantly outperforming supervised baselines in low-data regimes, rendering the model highly usable by downstream researchers even without dedicated finetuning.

By tackling the challenges of data heterogeneity, noise, and diverse instrumentation, AION-1 offers a promising paradigm for future multimodal foundation models beyond astronomy, setting the stage for a new era of large-scale, cross-domain scientific exploration.

## 2  Related Work

Multimodal foundation models have become a cornerstone of modern self-supervised learning [1, 20, 18, 36, 6, 60]. Indeed, recent advances like GPT-4V [1], Claude 3 [6], and LLaVA [36] have achieved human-level performance in visual reasoning, while models like Imagen [52] and Stable Diffusion [60] have enabled high-quality image generation from text. However, these models primarily rely on language to bridge modalities, which is often unavailable for scientific data. Recent work on early-fusion models, such as Chameleon [65], 4M [44], or PercieverIO [27], have demonstrated promising alternatives by learning mappings between modalities.

While these methodological advances in foundation models have transformed many fields, astronomy presents unique challenges, including heterogenous instruments, measurement protocols, and noise. As such, astronomy-specific efforts have emerged. For example, supervised pre-trained models like Zoobot [68] have leveraged 100M human annotations for galaxy morphology prediction obtained through extensive citizen science campaigns. Large-scale, self-supervised approaches trained on single-modal data have also emerged, including transformer-based models for `Gaia` stellar data [34], APOGEE spectra [32] and astronomical images [56] and contrastive approaches for astronomical images [25, 62, 61]. Finally, recent multimodal contrastive approaches have been introduced, starting with galaxy image-spectra pairs in [48] and followed by galaxy images and text [43] and time-series and photometry [80, 50].

Relative to these methods, AION-1 represents a significant advance in both scale and scope: it is the first effort to train multimodal models to billion-parameter scales and the first attempt to unify arbitrary modalites or different object types; in this case, 39 modalities across 200 million unique measurements spanning galaxies, stars, and quasars.

## 3  Universal Tokenization of Diverse Data

Tokenization in AION-1 transforms heterogeneous data into a unified, transformer-compatible representation. Astronomical datasets present two key challenges: the variety of data types (2D images, 1D spectra, scalar values) and the diversity of sources within each type (different telescopes, resolutions, and instrument formats). We address this through modality-specific tokenizers that provide intra-modality standardization; each modality uses a dedicated tokenizer capable of handling multiple instruments, ensuring aligned representations within each data type. Moreover, the need to train multiple tokenizers for a modality with multiple survey inputs is removed. We provide the full details of the tokenizers in Appendix C and the list of tokenized modalities AION-1 understands in Appendix J.

**Multiband imaging data.**  Galaxy images vary widely in resolution (physical pixel size), channel number, wavelength range, noise properties, and brightness across surveys. AION's image tokenizer addresses this diversity through a flexible channel-embedding scheme - adapted from [40] - that accommodates variable channel counts and embeds provenance information (e.g., originating telescope) for each channel. Built on a ResNet backbone adapted from [77] and using Finite-Scale

Quantization (FSQ) [42], our tokenizer enables a *single* model to ingest imaging data from multiple diverse observational pipelines.

The tokenizer is trained using an inverse-variance-weighted Gaussian negative log-likelihood (NLL) that leverages our prior knowledge of the noise properties in each image, as reported by the data-generation pipelines. The NLL is given by:

$$\mathcal{L}_{\text{NLL}} \ = \ \sum_i \frac{1}{2} \parallel \boldsymbol{\Sigma_i}^{-\frac{1}{2}} \left( \mathbf{x}_i - \text{Dec}_\theta(\text{Enc}_\phi(\mathbf{x}_i)) \right) \parallel_2^2 \tag{1}$$

where $\mathbf{x}_i$ is the input image, and $\boldsymbol{\Sigma}_i$ is the diagonal noise covariance of that image provided by the imaging pipeline, which accounts for background noise and shot noise from bright sources.

**Spectroscopic Data.** A single spectrum measures the wavelength dependent variation of light typically coming from a single object. Measurements from different spectrographs vary in amplitude, wavelength range, and resolution. AION's spectrum tokenizer normalizes and resamples spectra onto a shared latent wavelength grid, enabling joint processing of disparate instruments and objects, e.g. galaxies, stars, and quasars. Built on ConvNeXt V2 [74] with Look-up Free Quantization (LFQ) [79], it incorporates survey-specific noise variance through the same loss function used for images.

**Tabular/Scalar Data.** Scientific data often involve scalar measurements—both direct (e.g., size, flux) and derived (e.g., redshift). Instead of continuous embeddings [22], which struggle with large or variable dynamic ranges, AION-1 tokenizes scalars using uniform binning in their cumulative distribution functions (CDFs). This homogenizes the scalar distributions and minimizes quantization errors in the most important regions.

**Segmentation and Property Maps.** Beyond photometric images, other spatially resolved scalar fields (e.g., segmentation maps, physical property maps) are also of interest in galaxy astronomy. We thus include a tokenizer for normalized maps in $[0, 1]$, based on a convolutional architecture with FSQ quantization [42] and trained on grayscale galaxy images and segmentation maps.

**Bounding Ellipses.** We also include bounding ellipse catalogs for object localization. Each ellipse in these catalogs is characterized by a quintuple (spatial coordinates, ellipticity, radius), tokenized by mapping coordinates to the nearest pixel and quantizing elliptical attributes. Since object counts vary, we linearize these entries into a sequence [12], sorted by distance from the image center, providing a consistent reference and reducing ambiguities.

## 4 Multimodal Masked Modeling

AION-1 builds on recent early fusion multimodal models [65, 27, 21]. In particular, it adopts the scalable multimodal masked modeling scheme proposed in 4M [44, 7] to learn from heterogeneous data (e.g., spectra, images, scalars) by randomly masking tokenized inputs across all available modalities and reconstructing the masked content.

Concretely, let $\mathbf{X} = \{\mathbf{x}_1, \dots, \mathbf{x}_M\}$ be token sequences for $M$ modalities available for a training example. During training, two disjoint subsets of $\mathbf{X}$ are drawn: $\mathbf{x}^{\text{obs}}$ (observed) and $\mathbf{x}^{\text{tgt}}$ (target). Because these two subsets are sampled *across* the entire token pool, the model learns both intra- and cross-modal relationships in each training instance.

The loss for the model is then given by:

$$\mathcal{L}_{\text{4M}}(\theta) \ = \ -\sum_{t=1}^{N} \log \, p_\theta\big(\mathbf{x_t}^{tgt} \,\big|\, \mathbf{x}_t^{\text{obs}}\big), \tag{2}$$

where $p_\theta(\cdot \mid \mathbf{x}^{\text{obs}})$ is the categorical distribution over the predicted vocabulary, and $N$ is the output token budget.

### 4.1 Architecture

We adopt a Transformer-based encoder-decoder framework suitable for the multi-modal masked prediction (see Figure 1). Beyond a standard encoder-decoder architecture, we emphasize below

the modality specific embedding scheme needed at the input of both the encoder and decoder to implement our objective.

Concretely, each modality $i \in \{1, \ldots, M\}$ has its own token embedding $\text{Embed}_i(\cdot)$, a learnable modality embedding $\mathbf{m}_i$, and positional embedding $\mathbf{p}_t$ for token position $t$. Then, for an observed token from modality $i$, $x_t^i$, the full embedding is given by

$$\mathbf{e}_t^{(\text{enc})} \;=\; \text{Embed}_i(x_t^i) \;+\; \mathbf{m}_i \;+\; \mathbf{p}_t. \tag{3}$$

In the decoder, we feed information on the *target* we are querying tokens for, without providing their value:

$$\mathbf{e}_t^{(\text{dec})} \;=\; \mathbf{m}_i \;+\; \mathbf{p}_t, \tag{4}$$

omitting any direct lexical embedding $\text{Embed}_i(x_t)$.

In our implementation we use a different modality embedding $\mathbf{m}_i$ for *each modality* and *each source* to identify the unique combination of data type and associated provenance metadata. In other words, two astronomical images from two different instruments will have two different modality embeddings even though they are both images. This is to provide the model with important provenance information which implicitly encodes aspects of data quality and resolution of the observations.

## 4.2 Modality Masking Strategy

A key consideration is to select which tokens become *inputs* (observed) vs. *outputs* (predicted) for each modality during training. We find that the dirichlet sampling from the original 4M implementation is inefficient when dealing with modalities that vary widely in length, and therefrore results in a high frequency of mostly empty batches. Therefore, we follow a simplified approach:

**Input Token Budget**    We select a global input token budget $B$. To populate the budget, we randomly pick one modality, and then uniformly randomly select a number of tokens for inclusion from that modality. We then fill the remaining budget by uniformly sampling tokens from the other modalities.

**Output Token Budget.**    For the remaining unselected tokens, we choose the number of tokens to predict for each modality by sampling from a Beta distribution skewed toward zero, which draws down the number of output tokens per sample, aligning with the eventual distribution of output tokens under a cosine schedule iterative sampling (e.g., in MaskGIT-style), ensuring that inference-time usage patterns are well covered during training. Similar to the input, one modality is chosen to drawn an unconstrained number of tokens first, and the rest are filled by uniform random draws from the remaining modalities.

## 5 AION-1 Family of Models

**Dataset.**    AION-1 is pretrained on the publicly available data from the Multimodal Universe (hereafter MMU) [66], a large-scale dataset of ML-ready, multimodal astronomical data. We use five surveys: Hyper Suprime-Camera (HSC) [5] and Legacy Imaging Survey [16] for galaxy images; DESI [3] and SDSS [76] for high-resolution spectra and cosmic distances on galaxies and stars; and Gaia [13] for low-resolution spectra and precise photometric and astrometric measurements for stars in the Milky Way. We refer the reader to Appendix A for full details on the pretraining data. The relative contribution of each survey is illustrated in Figure 2a.

AION-1's pretraining emphasizes learning relationships between diverse observations of the same astronomical objects. Unlike 4M [44], which requires all modalities simultaneously, we use pairwise associations across surveys, which accommodates uneven associations.

**Training.**    We train three model versions - Base (300M), Large (800M), and XLarge (3B) - using the AdamW [37] optimizer ($\beta_1 = 0.9, \beta_2 = 0.95$, weight decay 0.05) for 205k steps with a global batch size of 8096. We use a linear warmup and cosine decay schedule, with a peak learning rate of $2 \times 10^{-4}$. Figure 2b shows how model size and data mixture (including or excluding Gaia) impact the evaluation loss.

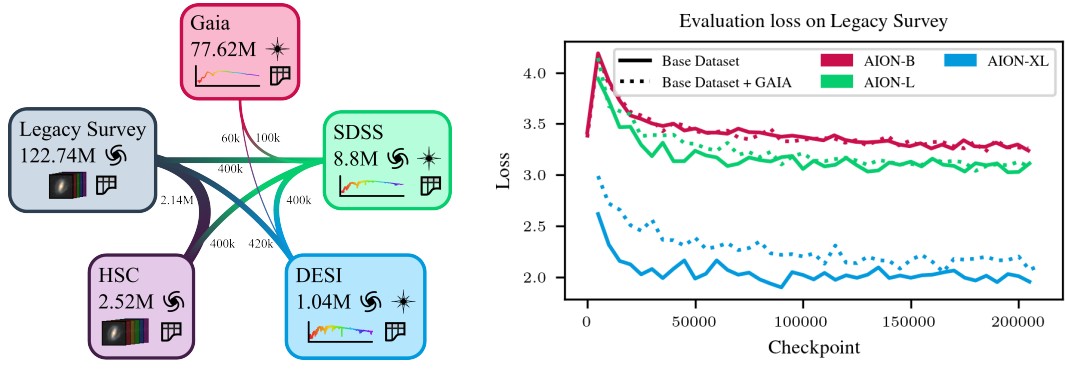

| (a) Training data mixture. | (b) Evaluation loss function |

Figure 2: **Left:** Illustration of the diverse data types used in pretraining. By pairing observations from different instruments along with object metadata, the model gains multimodal understanding. The small overlapping sample sizes—common in astrophysics—are a key motivation for AION-1. **Right:** AION-1 test losses on Legacy Survey [16] for various model sizes and dataset selection. It is interesting to note the change of losses across all model sizes when Gaia[13] - the only dataset that includes only stars (no galaxies) - is added.

## 6 Out-of-the-Box Capabilities

AION-1 enables a variety of generative tasks out-of-the-box, from data imputation to cross-survey translation. By representing the joint distribution over all modalities, AION-1 can draw conditional samples of any modality given partial observations, producing physically consistent reconstructions. We provide details on using AION-1 as a generative model in Appendix D.

**Cross-Modal Generation** AION-1 can conditionally generate high-dimensional modalities using the ROAR framework [44]. This enables scientific analyses like cross-survey translation or super-resolution. Here, we demonstrate AION-1's ability to generate high-resolution DESI spectra from coarse Gaia BP/RP coefficients that are 50-100× sparser than the DESI spectra. Fig. 3 demonstrates that the model accurately reconstructs line centers, widths, and amplitudes within narrow posterior uncertainty bands, enabling detailed abundance analysis from widely available low-resolution data. This capability is valuable given the significantly greater cost of performing high-resolution measurements like DESI and the much broader availability of low-resolution data like Gaia.

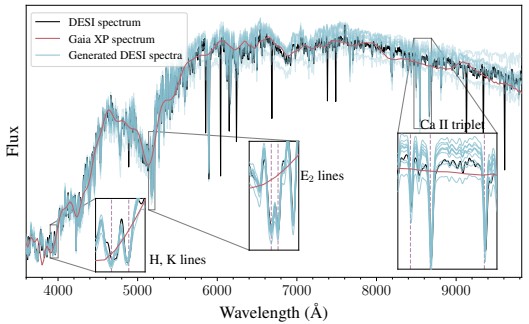

Figure 3: **Super-resolution**: AION-1 can generate high-resolution posterior spectra samples (blue) conditioned on low-resolution spectra input (red), closely matching the ground-truth high-resolution measurements (black) in line location, width, and amplitude.

**Scalar Posterior Estimation** AION-1 can also directly estimate any scalar value $z$ that is natively quantized during pretraining, allowing AION-1 to infer posterior distributions $p(z|\text{context})$. We demonstrate this capability on redshift, a key measure of cosmic distance. Figure 4 displays posterior samples for a representative galaxy under three increasingly informative contexts: (1) Legacy Survey photometry only, (2) Legacy Survey photometry and multi-band imaging, and (3) high-resolution DESI spectra. The posterior clearly contracts as more information is provided.

## 7 Evaluation with Downstream Scientific Workflows

AION-1 can produce physically meaningful, *modality-agnostic* embeddings out of the box, avoiding bespoke supervised pipelines. Because foundation models carry a pretraining prior, we use AION-1 as

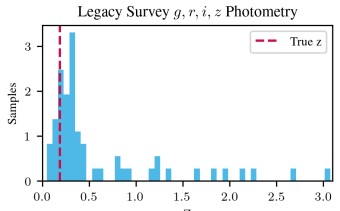 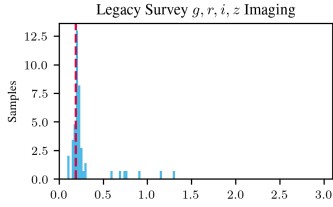 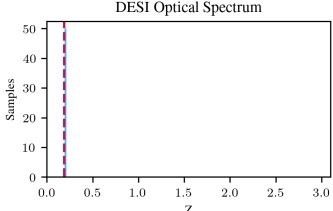

Figure 4: **Redshift Posterior Estimation.** Posterior samples for a single Legacy Survey galaxy under three conditioning scenarios: (left) broadband photometry only, (middle) photometry + imaging, and (right) full spectrum. The posterior contracts as richer information is provided.

a frozen feature extractor and perform lightweight task calibration: fit a small head on a representative calibration set (matching the downstream selection function) and calibrate outputs. At inference, we freeze the encoder and discard the decoder; given tokenized inputs, the encoder outputs a contextual sequence $\mathbf{Z} = \{\mathbf{z}_t\}_{t=1}^T$, $\mathbf{z}_t \in \mathbb{R}^d$, which we compress to an object vector $\mathbf{e} \in \mathbb{R}^d$ via mean pooling $\left(\mathbf{e} = \frac{1}{T}\sum_t \mathbf{z}_t\right)$ or learned attentive pooling $\left(\mathbf{e} = \mathrm{softmax}(\mathbf{QK}^\top/\sqrt{d})\mathbf{V}\right)$, with $\mathbf{Q}, \mathbf{K}, \mathbf{V}$ trainable. Multi-modality is native: concatenate tokens from any subset of modalities and pass the union through the frozen encoder—no extra fusion is required. This workflow preserves AION-1's representational power while letting researchers inject scientifically relevant priors.

## 7.1 Physical Property Estimation

Physical properties are used to describe fundamental aspects of astronomical objects, like the stellar mass of galaxies or temperature of stars. These are often derived using expensive, high-resolution measurements. As such, a common task in astronomy is to bypass this step and estimate these physical properties from low-resolution measurements directly with machine-learning models.

**Galaxy Properties** We predict five galaxy properties (redshift, $z$; stellar mass, $M_\star$; age, $t_{age}$; star-formation rate, *SFR*; metallicity, $Z_{\mathrm{Met}}$). We cross-match the SED-fit properties from PROVABGS [24] with the $\{g, r, i, z\}$ Legacy Survey imaging and DESI spectroscopy, yielding roughly 120,000 galaxies. We consider three input combinations: photometry, photometry + imaging, and photometry + imaging + spectra, and perform attentive pooling from the AION-1 embeddings as described above. We then fit a lightweight, two-layer multilayer perceptron (hidden size = 256) using a random 80/20 split, to map the input to the target parameters. We quantify performance with $R^2 = 1 - (\sum_j(y_j - \hat{y}_j)^2)/(\sum_j(y_j - \bar{y})^2)$, where $y_j$ are the ground-truth values, $\hat{y}_j$ the predicted values, and $\bar{y}$ the mean of the ground-truth sample. We include three supervised baselines informed by the literature: XGBoost on pho-

|  | $z$ | $M_\star$ | $t_{age}$ | $\log Z_{\mathrm{Met}}$ | sSFR |
|---|---|---|---|---|---|
| AION-1-B |  |  |  |  |  |
| *Ph* | 0.75 | 0.72 | 0.35 | 0.41 | 0.38 |
| *Ph+Im* | 0.93 | 0.89 | 0.45 | 0.49 | 0.64 |
| *Ph+Im+Sp* | 1.00 | 0.96 | 0.53 | 0.61 | 0.72 |
| AION-1-L |  |  |  |  |  |
| *Ph* | 0.76 | 0.73 | 0.36 | 0.41 | 0.39 |
| *Ph+Im* | 0.94 | 0.89 | 0.45 | 0.50 | 0.64 |
| *Ph+Im+Sp* | 1.00 | 0.96 | 0.53 | 0.62 | 0.73 |
| AION-1-XL |  |  |  |  |  |
| *Ph* | 0.79 | 0.76 | 0.31 | 0.38 | 0.48 |
| *Ph+Im* | 0.94 | 0.89 | 0.45 | 0.49 | 0.64 |
| *Ph+Im+Sp* | 0.99 | 0.95 | 0.53 | 0.62 | 0.73 |
| AstroCLIP [48] |  |  |  |  |  |
| *Im\** | 0.78 | 0.73 | 0.29 | 0.36 | 0.42 |
| *Sp* | 0.99 | 0.90 | 0.52 | 0.60 | 0.70 |
| DINOv2 [47] | 0.57 | 0.55 | 0.17 | 0.28 | 0.25 |
| Supervised |  |  |  |  |  |
| XGBoost (*Ph*) | 0.71 | 0.69 | 0.30 | 0.30 | 0.38 |
| ConvNeXt (*Im*) | 0.86 | 0.82 | 0.45 | 0.49 | 0.64 |
| Conv+Att (*Sp*) | 1.00 | 0.85 | 0.43 | 0.62 | 0.68 |

Table 1: **R$^2$ (↑) for galaxy property estimation.** Inputs are photometry (*Ph*), photometry + imaging (*Ph+Im*), and photometry + imaging + spectra (*Ph+Im+Sp*).

tometry, a ConvNeXt on imaging [66], and a CNN + attention network on spectra [41]. We also include results using the same MLP layer on embeddings produced by two self-supervised baselines, AstroCLIP [48] and DINOv2 [47][2].

---

[2]Note that AstroCLIP and DINOv2 can only ingest 3-channel imaging, so these are ran using correctly normalized $\{g, r, z\}$ Legacy Survey images while AION-1 is ran using the full $\{g, r, i, z\}$ channels.

|            | $\mathbf{T_{eff}}$ | $\log \mathbf{g}$ | $\mathbf{[Fe/H]}$ | $\mathbf{v_{mic}}$ |
|------------|------|-------|--------|-------|
| **AION-1-B** | | | | |
| *Ph* | 0.94 | 0.95 | 0.58 | 0.86 |
| *Ph+Plx+RA/Dec* | 0.94 | 0.95 | 0.70 | 0.87 |
| *Sp+Plx* | 0.99 | 0.98 | 0.94 | 0.89 |
| **AION-1-L** | | | | |
| *Ph* | 0.95 | 0.96 | 0.58 | 0.87 |
| *Ph+Plx+RA/Dec* | 0.95 | 0.96 | 0.71 | 0.88 |
| *Sp+Plx* | 0.99 | 0.98 | 0.94 | 0.89 |
| **AION-1-XL** | | | | |
| *Ph* | 0.92 | 0.94 | 0.56 | 0.85 |
| *Ph+Plx+RA/Dec* | 0.93 | 0.95 | 0.68 | 0.87 |
| *Sp+Plx* | 0.98 | 0.98 | 0.92 | 0.89 |
| Supervised | | | | |
| XGBoost (*Ph*) | 0.94 | 0.95 | 0.59 | 0.87 |
| ConvNeXt (*Sp*) | 0.99 | 0.98 | 0.95 | 0.89 |

Table 2: $R^2$ ($\uparrow$) **for stellar-label prediction.** Inputs are Gaia photometry (*Ph*), low-resolution Gaia XP spectra (*XP*), parallax (*Plx*), celestial coordinates (*RA/Dec*), and high-resolution DESI spectra (*Sp*).

**Stellar Properties** We also predict stellar parameters ($T_{eff}$; temperature, $\log g$; surface gravity, $v_{mic}$; microturbulent velocity, [Fe/H]; metallicity, [X/Fe]; elemental abundances). We cross-match *Gaia* DR3 data with DESI spectra and the associated derived properties from [80], yielding 240,000 stars. We consider three input combinations: photometry, photometry + parallax + sky position, and DESI spectra with parallax. We use the same adaptation head and training strategy as above and report $R^2$. We also include two supervised baselines from the literature: XGBoost on photometry [34] and ConvNeXt on raw spectra. Results are presented in Table 2.

In addition to these supervised baselines, we also compare performance to a state-of-the-art self-supervised baseline [34]. Specifically, we predict APOGEE-derived stellar parameters from *Gaia* XP spectral coefficients using cross-matched APOGEE-derived stellar parameters with *Gaia*, producing roughly 10,000 pairs. We feed only the first 32 BP coefficients and first 32 RP coefficients to AION-1 due to the fact that [34] has a context length of 64, and report results below. Note that [34] has been explicitly pretrained on this task, while AION-1 has never seen this task before; nonetheless, AION-1 outperforms.

| Model | $T_{eff}$ (K) | $\log g$ (dex) | [Fe/H] (dex) |
|-------|------|-------|--------|
| AION-B | 94.6 | 0.206 | 0.115 |
| Leung, et al. [34] | 99.1 | 0.229 | 0.143 |

Table 3: **Performance on APOGEE stellar property predictions from Gaia XP coefficients** ($\downarrow$), as measured by standard deviation of residuals. K is temperature units in Kelvin, and dex represents scatter on a logarithmic scale.

## 7.2 Learning from Semantic Human Labels

While some aspects of astronomical objects can be described by physical properties, in many cases they exhibit complex features that are not captured by physical models. We explore leveraging AION-1 to identify such features in galaxy images, given a limited set of human annotations.

**Galaxy Morphology Classification.** We consider here the problem of classifying galaxy images into ten distinct morphology classes (e.g. spiral arms, merging galaxies) defined by Galaxy Zoo 10 [GZ10; 69, 33]. We cross-match GZ10 with Legacy Survey images, yielding 8,000 galaxies. We produce embeddings from AION-1 using mean pooling as described above, and train a 2-layer MLP to classify galaxies from these embeddings. We present classification accuracy in Table 4. We see strong performance against a dedicated supervised convolutional model [64] trained from scratch, and competitive performance against a state-of-the-art model trained on two orders of magnitude more labeled data [70] that we adapt for this task with an MLP head on its frozen embeddings. We also demonstrate that AION-1 can be used to transfer learn from Legacy Survey morphologies to HSC morphologies in subsection H.4.

| Model | Accuracy (%) |
|-------|------|
| AION-1-B | 84.0 |
| AION-1-L | 87.2 |
| AION-1-XL | 86.5 |
| DINOv2 [47] | 71.4 |
| EfficientNet | 80.0 |
| ZooBot [69] | 89.6 |

Table 4: **Galaxy Morphology Classification Accuracy** ($\uparrow$) on Galaxy Zoo 10. AION-1, DINOv2 [47], and ZooBot [69] use an MLP head on frozen embeddings; EfficientNet-B3 [64] is trained end-to-end from scratch.

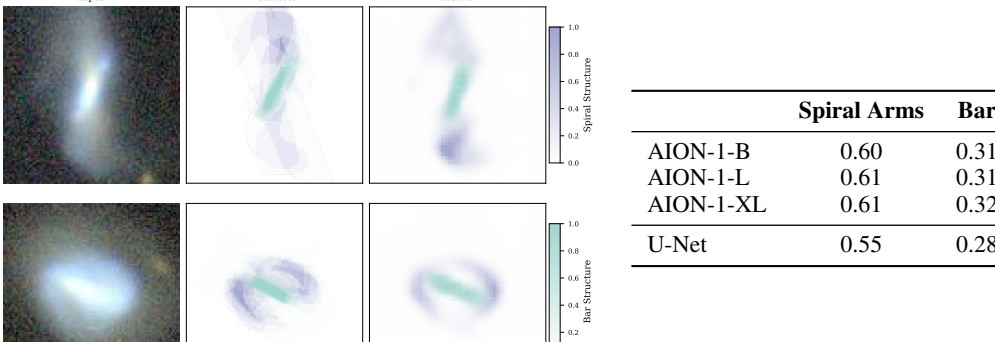

| | Spiral Arms | Bar |
|---|---|---|
| AION-1-B | 0.60 | 0.31 |
| AION-1-L | 0.61 | 0.31 |
| AION-1-XL | 0.61 | 0.32 |
| U-Net | 0.55 | 0.28 |

Figure 5: **Galaxy structure segmentation.** *Left*: Examples of galaxy image segmentation produced from the AION-1 embeddings with a lightweight convolutional neural network compared with the ground-truth true volunteer labels. *Right*: Mean IoU (↑) on the held-out test set.

**Semantic Segmentation of Galaxy Structures.** We also consider image segmentation based on human annotation of prominent galaxy structures obtained through the Galaxy Zoo 3D citizen science campaign [38] which we cross-match with Legacy Survey, yielding 2,800 galaxies. We train a simple convolutional upsampling head based on [31] on top of mean-pooled AION-1 embeddings to predict the segmentation maps. Training is performed using an 80/20 split. We compare our results to a simple fully convolutional U-Net baseline following architecture choices from [71]. We report Intersection over Union (IoU) between ground truth and crowd-annotated structural components. We also include in subsection G.3 a closely related task consisting in detecting regions of star formation in galaxy images.

## 7.3 Performance in Low-Data Regime

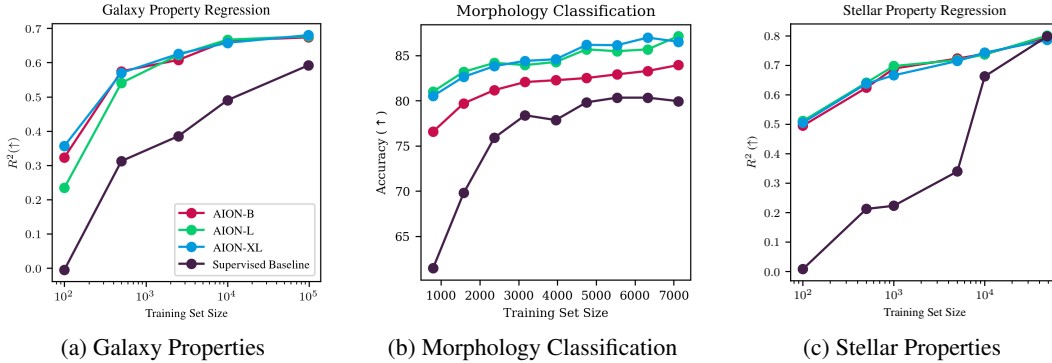

Figure 6: **Model performance vs downstream task training set size**. We regress **(a)** galaxy physical properties from images, **(b)** classify galaxy morphology, and **(c)** regress stellar properties from spectra. Supervised baselines are trained from scratch on the raw input data while AION-1 are results obtained by probing frozen encoder embeddings.

In astronomy, many key classes come with only a handful of reliable labels. As a result, approaches that perform well in low-data regimes are particularly valuable. To that end, we rerun the property prediction and morphology classification cases described above but artificially limit the training set size while keeping the test set fixed at 20%. For each experiment we retrain the lightweight MLP adapter as well as a corresponding supervised baseline. We note that AION-1's performance relative to supervised baselines is more impressive in the low data regime, where it matches or surpasses supervised models that require an order of magnitude more training data.

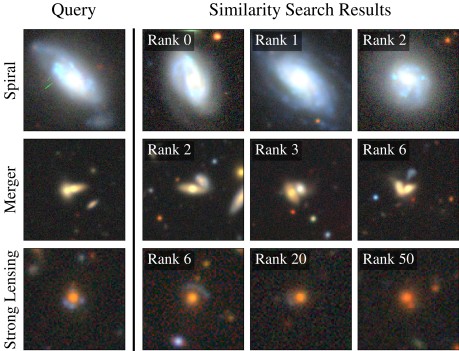

| | Spirals | Mergers | Lenses |
|---|---|---|---|
| AION-1-B | 0.632 | 0.281 | 0.012 |
| AION-1-L | 0.643 | 0.303 | 0.011 |
| AION-1-XL | 0.621 | 0.384 | 0.015 |
| AstroCLIP [48] | 0.602 | 0.248 | 0.006 |
| Stein et al. [62] | 0.590 | 0.340 | 0.007 |
| DINOv2 [47] | 0.477 | 0.060 | 0.003 |

Figure 7: **Galaxy Image Retrieval** for three astronomical classes of decreasing prevalence; spirals, mergers and strong lenses. *Left*: Example candidates for a given query image. *Right*: The nDCG@10 score for each of the classes averaged over all queries in the given dataset.

## 7.4 Similarity-Based Retrieval

Modern astronomical surveys gather immense volumes of data, however scientists often aim to identify a small set of rare phenomena that have outsized importance for understanding the Universe. For example, *strong gravitational lenses* are of particular interest in astronomy, as they provide unique constraints on dark matter and the expansion of the universe [54]. Despite their significance, these objects are notoriously hard to find, appearing in only a fraction of galaxies ($\sim 0.1\%$ for strong lenses), and thus lack large, annotated training sets for supervised methods.

AION-1 can address this challenge by searching for rare objects directly in the model's latent space. To illustrate this, we perform rare object retrieval for three types of rare objects; spirals and mergers from Galaxy Zoo DECaLS (25,000 and 700 out of 171,000 galaxies) [69] and strong lens candidates in the HSC footprint (700 out of 758,000 galaxies) [29], both of which we cross-match with the Legacy Survey data. For all cases, we embed all images in a shared latent space using the mean pooling strategy above, and then for all query objects, we use cosine similarity to rank all possible candidates and compute an nDCG@10 score (Appendix H.1), which we then average over the full query set; we provide full details of these datasets in the appendix. Figure 7 shows that AION-1 performs well on both Galaxy Zoo DECaLS objects like spirals (which make up 26% of the sample) and mergers (2%) [69] as well as strong gravitational lens candidates in the HSC footprint, which have just 0.1% occurrence: Appendix H.3). We compare AION-1's performance to leading self-supervised astronomy models [48, 62], as well as DINOv2 [47], all ran on the same dataset. We introduce strategies to further improve performance in Appendix H.2.

## 8 Discussion & Conclusion

AION-1 summarises heterogeneous astronomical observations in a single early-fusion backbone, yielding strong zero-shot performance and linear-probe accuracy that rivals (or in sparse-label regimes exceeds) task-specific models. By tokenising images, spectra and catalogues into a common sequence and training with masked-prediction objectives, the model learns cross-modal representations that support flexible data fusion, generative translation (e.g. sampling high-resolution spectra from low-resolution inputs) and efficient retrieval of rare phenomena, all without fine-tuning. These capabilities foreshadow a shift from bespoke, modality-specific pipelines toward reusable foundation models that streamline discovery and improve reproducibility across surveys.

**Limitations and outlook.** AION-1 inherits survey selection biases, lacks explicit temporal reasoning and remains computationally intensive to train, but none of these obstacles are fundamental. Continual pre-training on forthcoming facilities, time-aware tokenisers and advances in sparsity or distillation can broaden coverage and democratise use. More ambitiously, coupling physics-aware objectives could steer representations toward physically consistent manifolds while accelerating the curation of high-value labels. Even in its current form, AION-1 demonstrates that omnimodal foundation models are both feasible and transformative, providing a scalable template for cross-domain scientific AI.

# 9 Acknowledgments

We would like to acknowledge the support of the Simons Foundation and of Schmidt Sciences. This project was provided with computer and storage resources by GENCI at IDRIS thanks to the grant 2024-GC011015468 on the supercomputer Jean Zay's H100 partition. Additionally, some of the computations in this work were run at facilities supported by the Scientific Computing Core at the Flatiron Institute, a division of the Simons Foundation. Liam Parker also acknowledges support from the National Science Foundation Graduate Research Fellowship Program. Jeff Shen is supported by the Natural Sciences and Engineering Research Council of Canada (NSERC), funding reference number 587652. We would like to thank Andrew Engel, Marc Huertas-Company, Stephanie Juneau, Andy Morgan, and Mike Smith for their valuable feedback during beta testing and Sophie Barstein for her help with writing the corresponding blog post.

## 9.1 Data

### 9.1.1 Legacy Survey

The Legacy Surveys consist of three individual and complementary projects: the Dark Energy Camera Legacy Survey (DECaLS; Proposal ID #2014B-0404; PIs: David Schlegel and Arjun Dey), the Beijing-Arizona Sky Survey (BASS; NOAO Prop. ID #2015A-0801; PIs: Zhou Xu and Xiaohui Fan), and the Mayall z-band Legacy Survey (MzLS; Prop. ID #2016A-0453; PI: Arjun Dey). DECaLS, BASS and MzLS together include data obtained, respectively, at the Blanco telescope, Cerro Tololo Inter-American Observatory, NSF's NOIRLab; the Bok telescope, Steward Observatory, University of Arizona; and the Mayall telescope, Kitt Peak National Observatory, NOIRLab. Pipeline processing and analyses of the data were supported by NOIRLab and the Lawrence Berkeley National Laboratory (LBNL). The Legacy Surveys project is honored to be permitted to conduct astronomical research on Iolkam Du'ag (Kitt Peak), a mountain with particular significance to the Tohono O'odham Nation.

NOIRLab is operated by the Association of Universities for Research in Astronomy (AURA) under a cooperative agreement with the National Science Foundation. LBNL is managed by the Regents of the University of California under contract to the U.S. Department of Energy.

This project used data obtained with the Dark Energy Camera (DECam), which was constructed by the Dark Energy Survey (DES) collaboration. Funding for the DES Projects has been provided by the U.S. Department of Energy, the U.S. National Science Foundation, the Ministry of Science and Education of Spain, the Science and Technology Facilities Council of the United Kingdom, the Higher Education Funding Council for England, the National Center for Supercomputing Applications at the University of Illinois at Urbana-Champaign, the Kavli Institute of Cosmological Physics at the University of Chicago, Center for Cosmology and Astro-Particle Physics at the Ohio State University, the Mitchell Institute for Fundamental Physics and Astronomy at Texas A&M University, Financiadora de Estudos e Projetos, Fundacao Carlos Chagas Filho de Amparo, Financiadora de Estudos e Projetos, Fundacao Carlos Chagas Filho de Amparo a Pesquisa do Estado do Rio de Janeiro, Conselho Nacional de Desenvolvimento Cientifico e Tecnologico and the Ministerio da Ciencia, Tecnologia e Inovacao, the Deutsche Forschungsgemeinschaft and the Collaborating Institutions in the Dark Energy Survey. The Collaborating Institutions are Argonne National Laboratory, the University of California at Santa Cruz, the University of Cambridge, Centro de Investigaciones Energeticas, Medioambientales y Tecnologicas-Madrid, the University of Chicago, University College London, the DES-Brazil Consortium, the University of Edinburgh, the Eidgenossische Technische Hochschule (ETH) Zurich, Fermi National Accelerator Laboratory, the University of Illinois at Urbana-Champaign, the Institut de Ciencies de l'Espai (IEEC/CSIC), the Institut de Fisica d'Altes Energies, Lawrence Berkeley National Laboratory, the Ludwig Maximilians Universitat Munchen and the associated Excellence Cluster Universe, the University of Michigan, NSF's NOIRLab, the University of Nottingham, the Ohio State University, the University of Pennsylvania, the University of Portsmouth, SLAC National Accelerator Laboratory, Stanford University, the University of Sussex, and Texas A&M University.

BASS is a key project of the Telescope Access Program (TAP), which has been funded by the National Astronomical Observatories of China, the Chinese Academy of Sciences (the Strategic Priority Research Program "The Emergence of Cosmological Structures" Grant # XDB09000000), and the Special Fund for Astronomy from the Ministry of Finance. The BASS is also supported by the

External Cooperation Program of Chinese Academy of Sciences (Grant # 114A11KYSB20160057), and Chinese National Natural Science Foundation (Grant # 12120101003, # 11433005).

The Legacy Survey team makes use of data products from the Near-Earth Object Wide-field Infrared Survey Explorer (NEOWISE), which is a project of the Jet Propulsion Laboratory/California Institute of Technology. NEOWISE is funded by the National Aeronautics and Space Administration.

The Legacy Surveys imaging of the DESI footprint is supported by the Director, Office of Science, Office of High Energy Physics of the U.S. Department of Energy under Contract No. DE-AC02-05CH1123, by the National Energy Research Scientific Computing Center, a DOE Office of Science User Facility under the same contract; and by the U.S. National Science Foundation, Division of Astronomical Sciences under Contract No. AST-0950945 to NOAO.

### 9.1.2 Hyper Suprime-Cam

The Hyper Suprime-Cam (HSC) collaboration includes the astronomical communities of Japan and Taiwan, and Princeton University. The HSC instrumentation and software were developed by the National Astronomical Observatory of Japan (NAOJ), the Kavli Institute for the Physics and Mathematics of the Universe (Kavli IPMU), the University of Tokyo, the High Energy Accelerator Research Organization (KEK), the Academia Sinica Institute for Astronomy and Astrophysics in Taiwan (ASIAA), and Princeton University. Funding was contributed by the FIRST program from Japanese Cabinet Office, the Ministry of Education, Culture, Sports, Science and Technology (MEXT), the Japan Society for the Promotion of Science (JSPS), Japan Science and Technology Agency (JST), the Toray Science Foundation, NAOJ, Kavli IPMU, KEK, ASIAA, and Princeton University.

This paper makes use of software developed for the Large Synoptic Survey Telescope. We thank the LSST Project for making their code available as free software at http://dm.lsst.org

The Pan-STARRS1 Surveys (PS1) have been made possible through contributions of the Institute for Astronomy, the University of Hawaii, the Pan-STARRS Project Office, the Max-Planck Society and its participating institutes, the Max Planck Institute for Astronomy, Heidelberg and the Max Planck Institute for Extraterrestrial Physics, Garching, The Johns Hopkins University, Durham University, the University of Edinburgh, Queen's University Belfast, the Harvard-Smithsonian Center for Astrophysics, the Las Cumbres Observatory Global Telescope Network Incorporated, the National Central University of Taiwan, the Space Telescope Science Institute, the National Aeronautics and Space Administration under Grant No. NNX08AR22G issued through the Planetary Science Division of the NASA Science Mission Directorate, the National Science Foundation under Grant No. AST-1238877, the University of Maryland, and Eotvos Lorand University (ELTE) and the Los Alamos National Laboratory.

### 9.1.3 Dark Energy Spectroscopic Instrument

This research used data obtained with the Dark Energy Spectroscopic Instrument (DESI). DESI construction and operations is managed by the Lawrence Berkeley National Laboratory. This material is based upon work supported by the U.S. Department of Energy, Office of Science, Office of High-Energy Physics, under Contract No. DE–AC02–05CH11231, and by the National Energy Research Scientific Computing Center, a DOE Office of Science User Facility under the same contract. Additional support for DESI was provided by the U.S. National Science Foundation (NSF), Division of Astronomical Sciences under Contract No. AST-0950945 to the NSF's National Optical-Infrared Astronomy Research Laboratory; the Science and Technology Facilities Council of the United Kingdom; the Gordon and Betty Moore Foundation; the Heising-Simons Foundation; the French Alternative Energies and Atomic Energy Commission (CEA); the National Council of Science and Technology of Mexico (CONACYT); the Ministry of Science and Innovation of Spain (MICINN), and by the DESI Member Institutions: www.desi.lbl.gov/collaborating-institutions. The DESI collaboration is honored to be permitted to conduct scientific research on Iolkam Du'ag (Kitt Peak), a mountain with particular significance to the Tohono O'odham Nation. Any opinions, findings, and conclusions or recommendations expressed in this material are those of the author(s) and do not necessarily reflect the views of the U.S. National Science Foundation, the U.S. Department of Energy, or any of the listed funding agencies.

### 9.1.4 Sloan Digital Sky Survey

Funding for the Sloan Digital Sky Survey V has been provided by the Alfred P. Sloan Foundation, the Heising-Simons Foundation, the National Science Foundation, and the Participating Institutions. SDSS acknowledges support and resources from the Center for High-Performance Computing at the University of Utah. SDSS telescopes are located at Apache Point Observatory, funded by the Astrophysical Research Consortium and operated by New Mexico State University, and at Las Campanas Observatory, operated by the Carnegie Institution for Science. The SDSS web site is `www.sdss.org`.

SDSS is managed by the Astrophysical Research Consortium for the Participating Institutions of the SDSS Collaboration, including Caltech, The Carnegie Institution for Science, Chilean National Time Allocation Committee (CNTAC) ratified researchers, The Flatiron Institute, the Gotham Participation Group, Harvard University, Heidelberg University, The Johns Hopkins University, L'Ecole polytechnique fédérale de Lausanne (EPFL), Leibniz-Institut für Astrophysik Potsdam (AIP), Max-Planck-Institut für Astronomie (MPIA Heidelberg), Max-Planck-Institut für Extraterrestrische Physik (MPE), Nanjing University, National Astronomical Observatories of China (NAOC), New Mexico State University, The Ohio State University, Pennsylvania State University, Smithsonian Astrophysical Observatory, Space Telescope Science Institute (STScI), the Stellar Astrophysics Participation Group, Universidad Nacional Autónoma de México, University of Arizona, University of Colorado Boulder, University of Illinois at Urbana-Champaign, University of Toronto, University of Utah, University of Virginia, Yale University, and Yunnan University.

### 9.1.5 Gaia

This work has made use of data from the European Space Agency (ESA) mission *Gaia* (`https://www.cosmos.esa.int/gaia`), processed by the *Gaia* Data Processing and Analysis Consortium (DPAC, `https://www.cosmos.esa.int/web/gaia/dpac/consortium`). Funding for the DPAC has been provided by national institutions, in particular the institutions participating in the *Gaia* Multilateral Agreement. The Gaia data are open and free to use, provided credit is given to 'ESA/Gaia/DPAC'. If you use Gaia DR3 data in your research, please acknowledge it as above.

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

# A   Details on Pretraining Data

Below, we provide a brief overview of the details and data types from each survey, but refer the reader to [66] or the original source dataset paper for more exhaustive descriptions. We note that the per-survey magnitude/quality thresholds, footprint choices, Gaia XP availability, and our reciprocal cross-match together define the effective selection function of the pre-training corpus; this shapes which morphologies, redshifts, environments, and S/N regimes might be emphasized in AION-1's embeddings.

## A.1   Legacy Surveys

The DESI Legacy Imaging Surveys combine three wide-field programs on the Blanco, Mayall, and Bok telescopes, delivering uniform $\{g, r, i, z\}$ imaging over $\sim$20 000deg$^2$—roughly half the sky. The galaxies are imaged at a pixel scale of 0.262 arcsec. MMU provides $160 \times 160$ pixel postage stamp cut-outs centered on galaxies from the Data Release 10 [16], which we crop to $96 \times 96$ images; we use only objects in the Southern Galactic Cap, and retain objects with `mag_z` $< 21$ that pass the MMU quality cuts, corresponding to roughly 122 million galaxies.

**Modalities** Each object is packaged as: (1) calibrated $\{g, r, i, z\}$ integrated fluxes and their inverse-variance estimates, (2) mid-IR fluxes W1–W4 from *WISE*, (3) Milky-Way reddening $E(B-V)$, and (4) basic shape/size descriptors—the ellipticity components $(e_1, e_2)$ and circularised half-light radius $R_{\mathrm{eff}}$—derived from the Legacy `tractor` model fits.

## A.2   Hyper Suprime-Cam (HSC)

The Hyper Suprime–Cam Subaru Strategic Program delivers deep, high-resolution $\{g, r, i, z, y\}$ imaging over $\sim$1 200 deg$^2$. We use only the **wide** subset from PDR3 [5]. From the co–added `calexp` frames, MMU extracts $160 \times 160$-pixel cut-outs at a pixel scale of 0.162 arcsec centered on catalog sources, which we crop to $96 \times 96$, as with the Legacy Survey images. Objects are kept when they satisfy: `mag_i` $< 22.5$; at least three visits in every band ("full-depth full-colour"); and the standard HSC quality flags remove bright-star contamination, edge artefacts, saturation and unreliable `cmodel` photometry. The resulting sample contains roughly 2.5 million galaxies.

**Modalities** Each object is packaged as: (1) calibrated $\{g, r, i, z, y\}$ integrated fluxes and their inverse-variance estimates, (2) PSF-homogenised forced photometry in each band with extinction corrections, and (3) the moment-based SDSS shape tensor components $(\gamma_{11}, \gamma_{12}, \gamma_{22})$ computed by the HSC pipeline.

## A.3   Sloan Digital Sky Survey (SDSS)

The Sloan Digital Sky Survey (SDSS) [4] has obtained medium-resolution ($R \sim 2\,000$) optical spectra for millions of objects. We use the aggregated public optical spectra from the Legacy, SEGUE-1/2, BOSS and eBOSS programs[3], covering 3 650–10 400Å with resolutions of $R = \lambda/\Delta\lambda = 1,500$ at $3,800$Å and $R = 9,000$ at 9,000Å. We keep only primary, science–target spectra from plates flagged `PLATEQUALITY='good'`. This yields $\sim$4 million galaxies and stars.

**Modalities.** Each object is packaged as: (1) the optical spectrum, its inverse-variance estimates, and its wavelength and (2) the pipeline redshift.

## A.4   Dark Energy Spectroscopic Instrument (DESI)

The Dark Energy Spectroscopic Instrument [15] survey is collecting spectra for $\sim$40 million galaxies and quasars; MMU presently ingests the Early Data Release (EDR, 1% of the full survey) [14]. Each spectrum spans 3 600–9 800Å on a fixed 7,081-pixel grid at resolutions of $R = 2000$ at $3,600$Å and $R = 5,500$ at $9,000$Å and is distributed with flux, wavelength and inverse-variance arrays. We select spectra from the SV3 "one-percent" survey where `SV_PRIMARY` is true, `OBJTYPE='TGT'` and `COADD_FIBERSTATUS=0`, giving roughly 1 million galaxies, stars, and quasars.

---

[3]We note that the SDSS and BOSS instruments have different fiber aperture sizes, but in the present work we include them in the same dataset.

**Modalities.** Each object is packaged as: (1) the optical spectrum, its inverse-variance estimates, and its wavelength and (2) the pipeline redshift.

## A.5 Gaia

Gaia DR3 [13] provides low-resolution prism spectra from its blue (BP) and red (RP) photometers for 220 million Milky-Way sources in addition to precise astrometry and broad-band photometry. MMU stores each BP/RP spectrum as the 110 Gauss–Hermite coefficients released by the mission (55 BP + 55 RP), which can be resampled onto an 1 101-pixel wavelength grid via `GaiaXPy`. We include all DR3 objects that have a mean BP/RP spectrum, retaining the full set of associated photometric, astrometric and stellar-parameter metadata.

**Modalities.** Each object is packaged as: (1) The 110 BP/RP spectral coefficients, (2) four-parameter astrometry (sky coordinates and parallax), and (3) mean fluxes in the $G^4$, BP and RP bands.

## A.6 Cross-matching strategy

For each pair of surveys we perform a nearest-neighbour match within a 1 arcsec radius on the sky and keep only reciprocal matches. Every resulting match is materialised as its own dataset. Each object in these datasets therefore aggregates all modalities from both parent surveys so that a single file read yields a fully fused, multi-survey view of the same astrophysical object. During AION-1 pre-training we draw samples both from the individual survey datasets and from these cross-matched sets, as detailed in the next sections. We note that this procedure may preferentially retain bright, isolated, well-centered sources and may de-emphasize blended or offset systems, introducing a further selection effect on the joint training distribution (see subsec:limitations).

# B Details on AION-1 Models

We provide in Table 5 a description of the different configurations used for our suite of models.

Table 5: **Model size variants:** Following the choice of 4M model, we adopt the conventional T5 model sizes and naming schemes [49].

| Model | Encoder Blocks | Decoder Blocks | Model Dim | Num Heads | Total Params |
|---|---|---|---|---|---|
| AION-1-B | 12 | 12 | 768 | 12 | 300M |
| AION-1-L | 24 | 24 | 1024 | 16 | 800M |
| AION-1-XL | 24 | 24 | 2048 | 32 | 3B |

**Pretraining Details.** We adopt an input budget of 256 tokens, and output budget of 128 tokens for all our models during pretraining. All models are trained with bfloat16 mixed precision, and model distribution under PyTorch's Fully Sharded Data Parallel (FSDP) ZeRO-2 strategy. To achieve a batch size of 8192 in all cases, we train AION-1-B using 64 H100 GPUs for 1.5 days, AION-1-L using 100 H100 GPUs for 2.5 days, and 288 H100 GPUs for 3.5 days.

# C Details on Tokenizers

## C.1 Multi-Survey Image Tokenizer

### C.1.1 Preprocessing

Each input from an imaging survey provides (i) a per-band flux map $\mathbf{x}$, (ii) a pixel-wise inverse variance map $\Sigma$, and (iii) a per-pixel mask $\mathbf{m}$ for a given source. Our tokenizer ingests heterogeneous measurements drawn from both HSC (five filters $\{g, r, i, z, y\}$) and the Legacy Survey SGC (LS; four filters $\{g, r, i, z\}$). The two pipelines vary in central wavelength, pixel scale, zero-point, and noise. Therefore, we treat all bands from the surveys as distinct from eachother; i.e. $g$ from Legacy

---

[4]$G$ is the mission's very broad "white-light" band measured by the astrometric field CCDs.

Survey is treated as a different band than $g$ from HSC. We stack all distinct bands into a single *fixed* set of 9 channels (5 from HSC and 4 from LS), assigning a specific index to each channel. Next, we map every image into a 9-channel tensor, filling the subset of channels corresponding to that image's bands with flux values and setting any unused channels to zero; a binary mask $\mathbf{m}_c \in \{0,1\}^C$ tracks which bands are zeroed out. The result of this process is that all images drawn have the same dimension, and can be stacked into a single, heterogenous batch, while maintaining survey-specific provenance information. We then normalize the zero-points between surveys by rescaling HSC to the Legacy Survey zero-point of $22.5$ mag via $s = 10^{(\text{ZP}-22.5)/2.5}$, and multiply by the ratio of pixel scales. While these steps are not strictly necessary - as the bands are already separated above - we find that it helps with training stability. Finally, we apply an $\mathrm{arcsinh}$ normalization to the images to account for their high dynamic range, which we invert before computing the autoencoding loss. We find that adequate range compression is crucial for training stability.

### C.1.2 Architecture and quantization.

**Subsampled Linear Projection**    Given a batch of images, $\mathbf{x} \in \mathbb{R}^{B \times C \times H \times W}$ (batch by channels by height by width), we project it to a higher-dimensional space of size $\dim_{\text{out}} \approx 6C$ using

$$\hat{\mathbf{x}} \;=\; \alpha(\mathbf{m}_c)\big(\tilde{\mathbf{x}}W + b\big),$$

with learnable $W \in \mathbb{R}^{C \times \dim_{\text{out}}}$ and $b \in \mathbb{R}^{\dim_{\text{out}}}$. The scale factor $\alpha(\mathbf{m})$ keeps the feature norm invariant to missing channels. The projection is inverted after decoding. We introduce the subsampled linear projection to expand each image into a higher-dimensional embedding that disentangles survey-specific channel information while preserving feature norms even when some bands are missing.

**Autoencoder**    Once subsampled, we feed the output of the subsampled linear projection, which is now a $54$-dimensional image, into a ResNet-based autoencoder. Specifically, we use the MagViT architecture adapted from [77], in which we remove transformer blocks. The encoder therefore consists of 2 downsampling ResNet blocks, which reduce the dimensionality of the input image by a factor of 16, resulting in a laten space that is $24 \times 24 \times 512$; this is compressed to $d = 4$ dimensions before being fed to the quantizer. The output of the quantizer is then projected back to $512$ dimensions, before being upsampled in the decoder. In total, the ResNet-based autoencoder has roughly 50M learnable parameters.

**Quantizer**    At the bottleneck of the tokenizer, we quantize features into a discrete set of codes. We experiment with multiple approaches, but empirically, we find that Finite Scale Quantization [FSQ; 42] yields the best performance in terms of reconstruction fidelity and training stability. Further, to explore the trade-off between reconstruction loss and codebook utilization, we vary the codebook size from smaller (e.g., $2^4$) to larger (e.g., $2^{14}$) - following the recommended configurations in the FSQ paper - and observe that a size of $2^{12}$ offers a desirable balance: the reconstruction loss plateaus with larger codebooks, while code usage remains sufficiently high to avoid underfitting with smaller codebooks; see Figure 8 for reference. Consequently, our final configuration employs FSQ with codebook levels of $n_i = \{8,5,5,5\}$, equating to a rough size of $2^{12}$ codes.

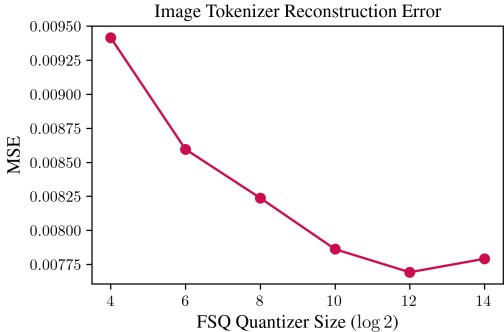

Figure 8: **Tokenizer Codebook Scaling**: We present the image tokenizer reconstruction error (MSE) as a function of FSQ quantizer size. We choose $2^{12}$ as our ultimate codebook size for images, as it reconstruction loss appears to plateau around this point.

### C.1.3 Loss Function and Per-band Weighting

The tokenizer is trained using an inverse-variance-weighted Gaussian negative log-likelihood (NLL) that leverages our prior knowledge of the noise properties in each image, as reported by the data-

| Original | Tokenizer Recon. | Tractor Recon. | Tokenizer Residual | Tractor Residual |

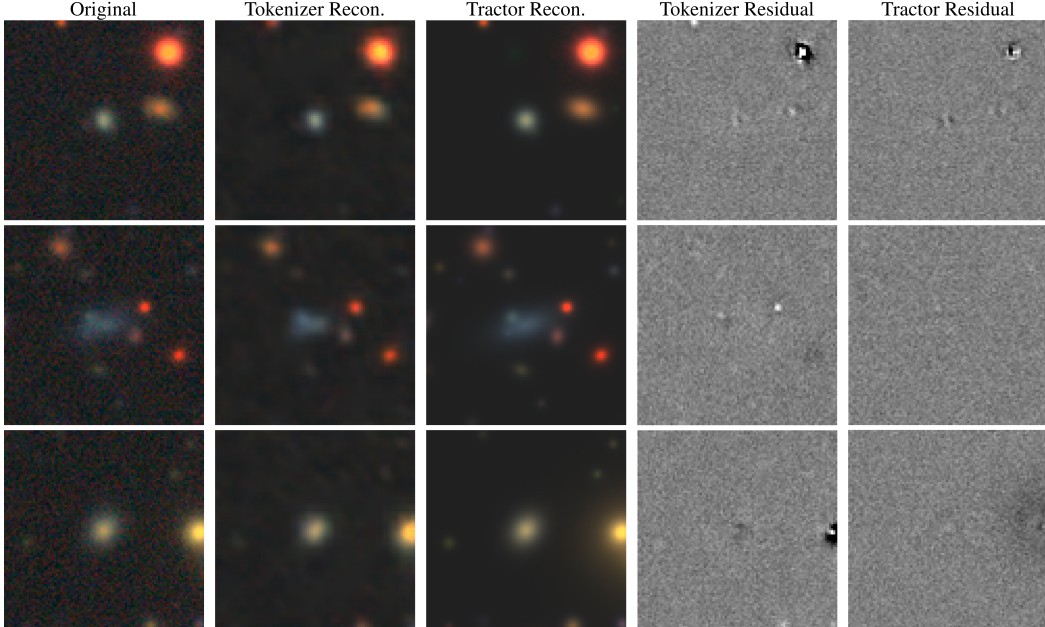

Figure 9: **Image Tokenizer Performance**: Reconstruction quality of the image tokenizer on three representative Legacy Survey images. The columns show, left to right, the original image, the reconstruction from the tokens, the reconstruction from the Legacy Survey tractor, the $r$-band residual for the tokenizer, and the $r$-band residual for the tractor.

generation pipelines. The NLL is given by:

$$\mathcal{L}_{\text{NLL}} \;=\; \sum_i \frac{1}{2} \parallel \mathbf{\Sigma_i}^{-\frac{1}{2}} \mathbf{m}_i \left( \mathbf{x}_i - \text{Dec}_\theta(\text{Enc}_\phi(\mathbf{x}_i)) \right) \parallel_2^2 \tag{5}$$

where $\mathbf{x}_i$ is the input image, $\mathbf{\Sigma}_i$ is the diagonal noise covariance provided by the imaging pipeline, accounting for background and shot noise from bright sources, and $\mathbf{m}_i$ is the survey pipeline mask which removes masked pixels in the image.

### C.1.4 Training Details

We train with the image tokenizer using the Adam [30] optimizer with a learning rate of $5 \times 10^{-4}$ on batches of 256 images, sampling LS:HSC at a 20:1 ratio to reflect the relative size of the two datasets. The learning rate is warmed up over 1k steps before being decayed for 400k steps using a cosine decay. Training converges in ~5 days on 4 × NVIDIA H100 GPUs, yielding a final reconstruction score of $\mathcal{L}_{\text{NLL}} = 0.00775$. We show some representative samples of the tokenizer's reconstruction quality in Figure 9, and include reconstructions from the Legacy Survey pipeline tractor for comparison.

### C.2 Multi-Survey Spectrum Tokenizer

### C.2.1 Preprocessing

Each input spectrum provides (i) observed-frame flux density per unit wavelength $\mathbf{f}(\lambda)$, (ii) inverse standard deviation $\mathbf{istd}(\lambda)$, and (iii) a per-pixel mask $\mathbf{m}(\lambda)$. Our tokenizer ingests heterogeneous measurements drawn from both DESI and SDSS. For each survey, we compute a robust median flux $\tilde{\mathbf{f}}$ (ignoring masked pixels), use a $\log_{10}$ range compression, and quantize it with a 1-D scalar tokenizer (codebook size = 1024). We then normalize $\mathbf{f}$ and $\mathbf{istd}$ by $\tilde{\mathbf{f}}$ and stack them into a 2-channel array. The array is linearly interpolated onto a fixed latent grid of 8704 points covering 3500–10462.4Å at 0.8Å spacing; this common grid is then shared between SDSS and DESI, and removes survey-specific wavelength/dispersion differences.

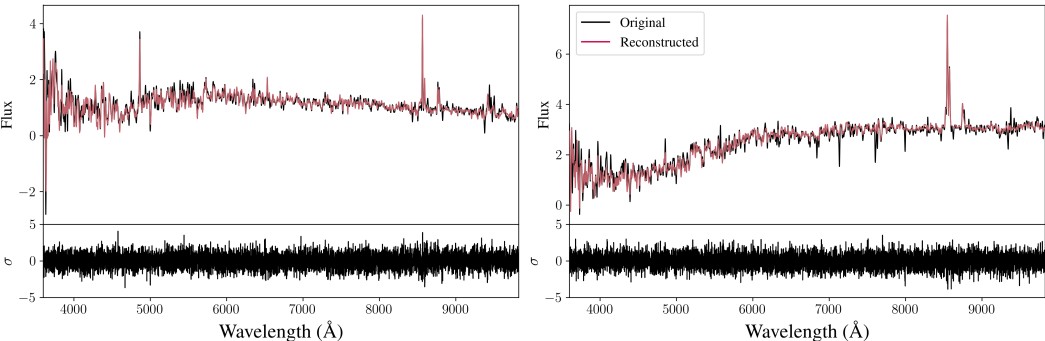

Figure 10: **Spectrum Tokenizer Performance**: Reconstruction quality of the spectrum tokenizer on two representative DESI EDR spectra. Top panels show the normalized flux as a function of wavelength, with original input (black) and reconstructed output (red) overlaid. Bottom panels show the residuals in units of the input uncertainty ($\sigma$).

### C.2.2 Architecture & Quantization

**Autoencoder** The stacked spectrum flux and inverse standard deviation $\mathbf{x} \in \mathbb{R}^{B \times 2 \times 8704}$ is encoded by a 4-stage ConvNeXt-V2 backbone [75], consisting of an initial downsampling stack composed of a $4 \times 4$ convolution and LayerNorm, followed by three downsampling stacks of $2 \times 2$ convolutions and LayerNorms. Each of the four downsampling stacks is followed by multiple ConvNeXt V2 processing blocks. This compresses the spectrum into a $273 \times 512$ latent space, which is then further downsampled to 10 dimension before being fed to the quantizer; this dimensionality is chosen to conform to the $2^{10}$ codebook size used by the quantizer. Like with the image, these steps are inverted during the decoding part of the autoencoder.

**Quantizer** We use a Look-up-Free Quantizer [LFQ; 78] with an embedding dimension of ten (equating to a codebook size of 1024 codes) to convert the latent sequence into discrete codes. Contrary to the images, we find here here that LFQ quantization slightly outperforms FSQ for the spectrum.

### C.2.3 Losses

For each spectrum we project the decoder output back to its native wavelength grid and apply three losses:

1. **Flux likelihood.** Gaussian NLL weighted by inverse variance $w(\lambda)$, identical to Eq. (5).

2. **Mask accuracy.** Binary cross-entropy between the predicted reliability map $\hat{\mathbf{m}}(\lambda)$ and the ground-truth mask $\mathbf{m}(\lambda)$.

3. **Commitment.** LFQ commitment loss with weight $\beta_{\mathrm{q}} = 0.25$.

### C.2.4 Training Details

We train with using the AdamW optimizer with a constant $10^{-4}$ learning rate, a 0.1 weight decay penalty, and a global batch size of 128. Training for 215 k steps ($\sim$24 hours on 4 × NVIDIA H100) yields a token reconstruction $R^2 = 0.994$ and a mean mask AUC of 0.92. Reconstruction quality on two representative spectra from DESI are shown in Figure 10.

### C.3 Scalar Tokenizer

While the scalar values could be quantized directly, equal width binning directly in the data space would lead to an uneven probability mass assignment and potentially imbalance training. Therefore, we map every scalar value to a unit normal Gaussian before quantization. To that end, we first need

to tabulate the empirical cumulative distribution function $F_x$ on the training set[5]. Once tabulated, we map a scalar value $x_i$ to a standard normal variate via

$$z_i = \Phi^{-1}\big(F_x(x_i)\big), \tag{6}$$

where $\Phi^{-1}$ is the inverse CDF of $\mathcal{N}(0,1)$. Because $z \sim \mathcal{N}(0,1)$, equal-width binning in $z$-space allocates the same probability mass to every bin, automatically adapting to long tails or sharp peaks in the original distribution. Each Gaussianised scalar $z_i$ is quantised independently with an FSQ codebook of $K = 1024$ centroids. Centroids are fixed a priori: we place them at the $K$ equally spaced quantiles of the standard normal, i.e. $c_k = \Phi^{-1}\big((k - \frac{1}{2})/K\big)$. No parameters are learned and no loss is required. To recover an approximate scalar value $\hat{x}_i$ from its token $c_i$,

$$\hat{x}_i = F_x^{-1}\big(\Phi(c_i)\big).$$

With $K = 1024$ bins the median absolute reconstruction error is below typical measurement uncertainties, ensuring that tokenisation fidelity is sufficient for downstream tasks while keeping the representation compact and parameter-free. Note that for some of the scalars with large dynamic ranges, we also apply a $\log_{10}$ or $\mathrm{arcsinh}$ transform before CDF mapping and tokenization. We apply the scalar tokenizer to the following scalars from each survey:

- **Legacy Survey**: $\{g, r, i, z\}$ fluxes, WISE W1-W4 fluxes, $E(B - V)$ extinction, ellipticity components $(e_1, e_2)$, circularized half-light radius $R_{\mathrm{eff}}$.
- **HSC**: $\{g, r, i, z, y\}$ fluxes, shape tensor components.
- **SDSS & DESI**: pipeline-reported redshift $(z)$.
- **Gaia**: 110 BP/RP coefficients, parallax, sky coordinates $(ra, dec)$, $G$, BP, RP fluxes.

### C.4 Scalar Field Tokenizer

In addition to images, we included an additional tokenizer specialized for scalar maps with values in $[0, 1]$. This tokenizer is particularly adapted to handle segmentation maps, but could also be used to generate any property map scaled between 0 and 1, such as Star Formation Rate maps derived from Integral Field Spectroscopy.

#### C.4.1 Data & Preprocessing

The scalar field tokenizer was trained to autoencode a mixture of 5 categories of normalized single-channel images derived from Legacy Survey photometry: RGB cutouts converted to grayscale; individual red, green, and blue channels from the RGB cutouts; and an 'object mask' indicating the silhouettes of sources detected in each cutout. The object mask is generated from the Tractor model photometry included in the Legacy Survey data release. The Tractor classifies each detected source as one of 5 morphological types and fits a corresponding elliptical surface brightness model to the light emitted by the source. After fitting, parameters can be extracted from the surface brightness profile to define a centered ellipse enclosing 50% of the total emission from a given source. We generate an object mask for each cutout by painting such ellipses onto a null background for all sources detected in the cutout. The ellipses are filled with a constant value selected from $\{0.2, 0.4, 0.6, 0.8, 1.0\}$ which corresponds to the morphological type.

#### C.4.2 Architecture & Quantization

**Autoencoder**  We base our architecture on VQ-VAE [67]; the encoder is comprised of a stack of 3 convolutional downsampling layers followed by 2 residual blocks, and this arrangement is mirrored in the decoder (with upsampling transpose convolutions replacing the downsampling convolutions). The downsampling convolutions have kernel size 4, stride 2, padding 1, and 128 / 256 / 512 kernels, respectively. Each residual block consists of a sequence of 2 convolutional layers separated by a batch norm layer, where the convolutions have kernel size 3 / 1, stride 1, and padding 1 / 0, respectively. All layers are ReLU-activated.

---

[5]We estimate $F_x$ with a fixed-size *reservoir* ($N \sim 10^6$ samples) maintained online during data streaming. Reservoir sampling (Algorithm R) produces an unbiased CDF while keeping memory $\mathcal{O}(N)$, independent of the full catalog size.

**Quantizer** We use a Finite Scale Quantizer [42] to quantize 4-dimensional codes with $n_{\text{dim}} = \{8, 5, 5, 5\}$ discrete levels available in the respective dimensions, yielding a codebook size of 1000.

### C.4.3 Training Details

This tokenizer was optimized under a mean squared error objective using the AdamW optimizer with the weight decay parameter set to 0.01. The model was trained with batch size 256 for 114,000 steps at a base learning rate of $10^{-4}$. The base learning rate was modified by a linear warmup phase in the first 1,000 steps and a cosine decay over the final 72,000 steps. The model weights were updated as an exponential moving average of previous values with a decay parameter of 0.9999. Under these conditions, the loss converged to $\mathcal{L}_{\text{MSE}} \approx 0.0017$.

## D  Details on Generative Capabilities

AION-1 is a generative model: once pretrained, it represents the joint distribution of all 39 tokenised modalities. At inference time we can therefore draw posterior samples of any modality in the training set by passing the appropriate query tokens and iteratively resampling them conditioned on the visible context. To generate these posteriors, we perform the following steps.

First, we pass the query modality through its appropriate tokenizer, producing a set of *input tokens* $\mathbf{x}^{\text{in}} = (x_1, \ldots, x_{N_{\text{in}}})$. These are passed to the AION-1 encoder, while the decoder receives a sequence of *query tokens* $\mathbf{x}^{\text{qry}} = (x_{N_{\text{in}}+1}, \ldots, x_N)$ whose values are to be inferred. At test-time we need samples from

$$p_\theta\big(\mathbf{x}^{\text{qry}} \,\big|\, \mathbf{x}^{\text{in}}\big) \;=\; \prod_{j \in \mathcal{Q}} p_\theta\big(x_j \,\big|\, \mathbf{x}^{\text{in}}\big), \tag{7}$$

where $\mathcal{Q}$ indexes the query positions and $p_\theta$ is the categorical distribution produced by the frozen decoder. To perform this sampling, we follow the ROAR generation scheme introduced in 4M [44]: at each iteration $t$ we

1. Draw a fresh *random permutation* $\pi_t : \mathcal{Q}_t \to \mathcal{Q}_t$ of the still-unknown query indices $\mathcal{Q}_t$;

2. Reveal the first $\rho_t = \lfloor r^t |\mathcal{Q}_t| \rfloor$ positions of this permutation,

$$\mathcal{S}_t \;=\; \big\{\pi_t(1), \ldots, \pi_t(\rho_t)\big\};$$

3. Sample those tokens once from the model,

$$x_j^{(t)} \sim p_\theta\big(\cdot \,\big|\, \mathbf{x}^{\text{in}} \cup \mathbf{x}^{(t-1)}_{\mathcal{Q}_{t-1} \setminus \mathcal{S}_t}\big), \quad j \in \mathcal{S}_t;$$

4. Promote them to inputs: $\mathbf{x}^{\text{in}} \leftarrow \mathbf{x}^{\text{in}} \cup \{x_j^{(t)}\}_{j \in \mathcal{S}_t}$ and update $\mathcal{Q}_{t+1} = \mathcal{Q}_t \setminus \mathcal{S}_t$.

With a decay factor $r \in [0, 1)$ the number of unresolved tokens drops exponentially, so the full sample is generated in $T = \mathcal{O}(\log |\mathcal{Q}|)$ decoder calls. After sampling is complete, every query token is routed back through its modality-specific tokenizer to recover the original data representation. Repeating the entire ROAR loop $M$ times with a non-zero sampling temperature $\tau > 0$ yields $M$ i.i.d. draws $\{\hat{\mathbf{x}}^{(k)}\}_{k=1}^{M}$ from the conditional distribution in Equation 7.

Importantly, we note that these draws are plausibility samples from the decoder's categorical outputs under an iterative reveal schedule; they are not guaranteed to be well-calibrated joint posteriors for sequences of tokens longer than a single token. We discuss this limitation in further detail in subsec:limitations.

## E  Details on Galaxy and Stellar Parameter Inference

**Data (Galaxy Parameter Inference).**  For inferring galaxy parameters, we use the data from the PRObabilistic Value-Added Bright Galaxy Survey (PROVABGS) [24], which provides derived physical properties from galaxy photometry and spectroscopy using a Bayesian Spectral Energy Density (SED) physical model. In particular, we extract:

- $z$: Redshift of the galaxy

- $Z_{\mathrm{met}}$: Metallicity
- $M_*$: Stellar mass
- $t_{\mathrm{age}}$: Stellar population age
- SFR: Star formation rate

We cross-match these galaxies with the Legacy Survey [16] imaging and photometry in the southern hemisphere and DESI spectroscopy [2]. Then, we apply several quality cuts to ensure reliable parameter estimates. In particular, we remove any objects with $M_* < 0$ as well as objects with any unphysical photometric magnitudes ($m < 0$). After these cuts, we arrive at a relatively clean sample of $\approx 100,000$ galaxies suitable for testing and validating our inference pipeline.

To accommodate the large dynamic range in certain parameters, we take the logarithm of both $Z_{\mathrm{met}}$ and $M_*$. We also convert SFR into the specific star formation rate (sSFR), given by

$$\mathrm{sSFR} \;=\; \log \frac{\mathrm{SFR}}{M_*}, \tag{8}$$

which is often more insightful for characterizing the relative growth of stellar mass in the galaxy.

**Data (Stellar Parameter Inference).**   The *Gaia* dataset that we train AION-1 on is a subset of the available data in *Gaia* DR3. We cross-match this subset with DESI to produce the sample for which we evaluate stellar parameter inference. The training set and validation sets consist of the cross-matched stars belonging to the training and validation healpixes during pretraining to ensure that the validation is performed on stars that AION-1 has not seen during pretraining.

While the MMU dataset for DESI provides stellar spectra, we turn to the catalog of [80] (hereafter Z24) for stellar parameters. Z24 uses a data-driven method with regularization from physical models to provides estimates for basic stellar parameters like $T_{\mathrm{eff}}$, $\log g$, and [Fe/H], as well as for various abundances [X/Fe] from DESI spectra.

**Models.**   We perform parameter inference by training linear models on top of AION-1 embeddings. Given some data inputs, we first tokenize them, then extract the corresponding embeddings from the pretrained and frozen AION-1 encoder. We experiment with both mean-pooling and cross-attention to compress the sequence dimension of the embeddings before applying a linear projection layer to the channel dimension to produce estimates for each quantity of interest.

**Data Fusion.**   We train all downstream models on various input combinations as a demonstration of data fusion. In particular, for galaxies we train downstream models on galaxy photometry, galaxy photometry and imaging, and galaxy photometry, imaging and spectroscopy. For stars: we train downstream models on stellar photometry, stellar photometry and low resolution spectroscopy, and low resolution spectroscopy and parallax.

**Supervised Baselines**   In the following section we detail the dedicated, supervised baselines used for galaxy and stellar property prediction.

We train two baseline models for stellar property prediction:

- **ConvNeXt Regressor on Raw Spectra.** This model uses multiple stacks of alternating ConvNeXt [74] processing blocks and downsampling blocks—identical in architecture to the encoder used for AION-1 spectrum tokenization—followed by attention pooling and a final linear projection. It is trained directly on the raw, pixel-level stellar spectra and noise estimates.

- **XGBoost on Token Representations.** We train an XGBoost regressor on the mean-pooled tokens produced by the AION-1 spectrum tokenizer. Because these tokens are already a compressed, high-level representation of the spectra, the XGBoost model's task is, in principle, simpler than that of the ConvNeXt regressor, which must simultaneously learn to extract features and perform the final prediction.

For galaxy property prediction, we train three different baseline models:

- **XGBoost on Photometry.** We train an XGBoost regressor directly on the photometric measurements of galaxies (e.g., magnitudes or fluxes in various bands).

- **ConvNeXt-Tiny on Images.** As a second baseline, we adopt the ConvNeXt-tiny architecture trained on galaxy images. This approach matches the baseline provided in the *Multi-Modal Universe* (MMU) framework.

- **Convolutional + Attention Network on Spectra.** Inspired by the method of [41] and used as a baseline in [48], we train a network that combines convolutional layers with attention-based pooling on the galaxy spectra.

**Self-Supervised Baselines.** We also use a number of state-of-the-art self-supervised baselines from the literature. For galaxies, we use the following models:

- **AstroCLIP** [48], a previous state-of-the-art multimodal foundation model for galaxies; we follow the authors' recommended protocol, extracting frozen embeddings from the CLIP image encoder and training a lightweight MLP ontop of the embeddings. Note that AstroCLIP was trained on Legacy Survey $\{g, r, z\}$ cut-outs only, so in our setting it has access to one fewer band ($i$) than AION-1.

- **DINOv2** [47] represents a widely used vision model; we feed RGB-converted $\{g, r, z\}$ images to the ViT-g/14 backbone and again attach the same MLP probe.

For stars, we compare performance against a current state-of-the-art baseline from [34], who developed a Transformer-based foundation model for stellar data. More specifically, the task is to predict APOGEE-derived stellar parameters - namely $T_{\text{eff}}$, $\log g$, and [Fe/H] - from Gaia XP spectral coefficients. We use the same data as [34], and cross-match APOGEE-derived stellar parameters with the MMU Gaia data, producing a set of roughly $\sim 10,000$ APOGEE parameter-Gaia XP spectral pairs. We feed as input to both AION-1 and the [34] model only the first 32 BP coefficients and first 32 RP coefficients due to the fact that the [34] model only has a context length of 64; we artificially handicap AION - which does not have this restriction - in order to perform a fair comparison. We note here that [34] has been explicitly given APOGEE-derived stellar parameters and Gaia XP coefficients during its pretraining stage, and so this task is one that it has effectively been trained for. On the other hand, the pretraining dataset for AION-B does not contain APOGEE data, nor any other stellar parameters; we simply train a simple linear projection layer with cross-attention pooling on 5000 paired examples, and leave the weights of the AION model itself frozen.

**Performance and Scaling.** We present below an extended set of results on stellar and galaxy parameter inference.

In Figure 12 we show the performance of AION-1-B, AION-1-L, and AION-1-XL, as well of the two baselines, on the prediction of individual properties as a function of training set size. We find that in general, AION-1 outperforms the baselines across the board, with the XGBoost baseline typically being slightly worse and the ConvNet baseline being significantly worse, especially in the low-data regime where only 100 training samples are available. However, while the ConvNet baseline performs very poorly with few samples, its performance improves with training set size up until the full training sample of $\sim 50,000$ examples, at which point it performs essentially identically to—and in some cases better than—AION-1-B/L/XL and the XGB baseline. It is possible that the regression task is not particularly difficult, and thus model performance saturates early; with AION-1-B performing essentially the same as AION-1-L, we find that scaling up to a large model provides little benefit **for this task**.

Figure 11 shows that overall, independent of input modalities, cross-attention pooling (solid lines) significantly outperforms mean-pooling (dashed lines).

# F   Details on Morphology Classification

**Data.** We construct the downstream sample by cross-matching the Galaxy Zoo 10 catalog [69, 33, GZ10;] with the Legacy Survey DR10 imaging footprint, yielding $\sim 8,000$ galaxies with $\{g, r, i, z\}$ cutouts.

|  | z | | $M_\star$ | | $t_{age}$ | | $Z_{Met}$ | | SFR | |
| --- | --- | --- | --- | --- | --- | --- | --- | --- | --- | --- |
|  | Mean | Attention | Mean | Attention | Mean | Attention | Mean | Attention | Mean | Attention |
| AION-1-B | | | | | | | | | | |
| *Ph* | 0.736 | 0.754 | 0.714 | 0.720 | 0.350 | 0.353 | 0.409 | 0.412 | 0.410 | 0.378 |
| *Ph+Im* | 0.910 | 0.934 | 0.857 | 0.886 | 0.394 | 0.445 | 0.453 | 0.490 | 0.611 | 0.637 |
| *Ph+Im+Sp* | 0.779 | 0.995 | 0.739 | 0.956 | 0.258 | 0.532 | 0.365 | 0.610 | 0.439 | 0.720 |
| AION-1-L | | | | | | | | | | |
| *Ph* | 0.659 | 0.761 | 0.630 | 0.734 | 0.268 | 0.357 | 0.302 | 0.411 | 0.228 | 0.387 |
| *Ph+Im* | 0.922 | 0.940 | 0.870 | 0.889 | 0.412 | 0.454 | 0.460 | 0.496 | 0.621 | 0.642 |
| *Ph+Im+Sp* | 0.800 | 0.995 | 0.760 | 0.955 | 0.276 | 0.534 | 0.375 | 0.620 | 0.461 | 0.727 |
| AION-1-XL | | | | | | | | | | |
| *Ph* | 0.679 | 0.792 | 0.647 | 0.757 | 0.266 | 0.314 | 0.318 | 0.379 | 0.240 | 0.475 |
| *Ph+Im* | 0.910 | 0.940 | 0.857 | 0.888 | 0.394 | 0.450 | 0.439 | 0.490 | 0.610 | 0.644 |
| *Ph+Im+Sp* | 0.795 | 0.992 | 0.759 | 0.947 | 0.273 | 0.534 | 0.374 | 0.621 | 0.454 | 0.731 |
|  | Supervised | | Supervised | | Supervised | | Supervised | | Supervised | |
| *Ph*[1] | 0.708 | | 0.692 | | 0.301 | | 0.301 | | 0.377 | |
| *Im*[2] | 0.864 | | 0.821 | | 0.445 | | 0.489 | | 0.638 | |
| *Sp*[3] | 0.998 | | 0.852 | | 0.433 | | 0.621 | | 0.675 | |

Table 6: $R^2$ (↑) **for galaxy property estimation.** Inputs to the model are: photometry (*Ph*), photometry and imaging (*Ph+Im*), and photometry, imaging, and spectra (*Ph+Im+Sp*). Mean implies taking the average over AION-1 embeddings followed by a linear projection head, while Attention implies training a cross-attention layer with a linear projection head on the full set of embeddings. Supervised models are: [1]XGBoost, [2]ConvNext, [3]Convolution + Attention Network. All models are trained on $\sim 100,000$ examples.

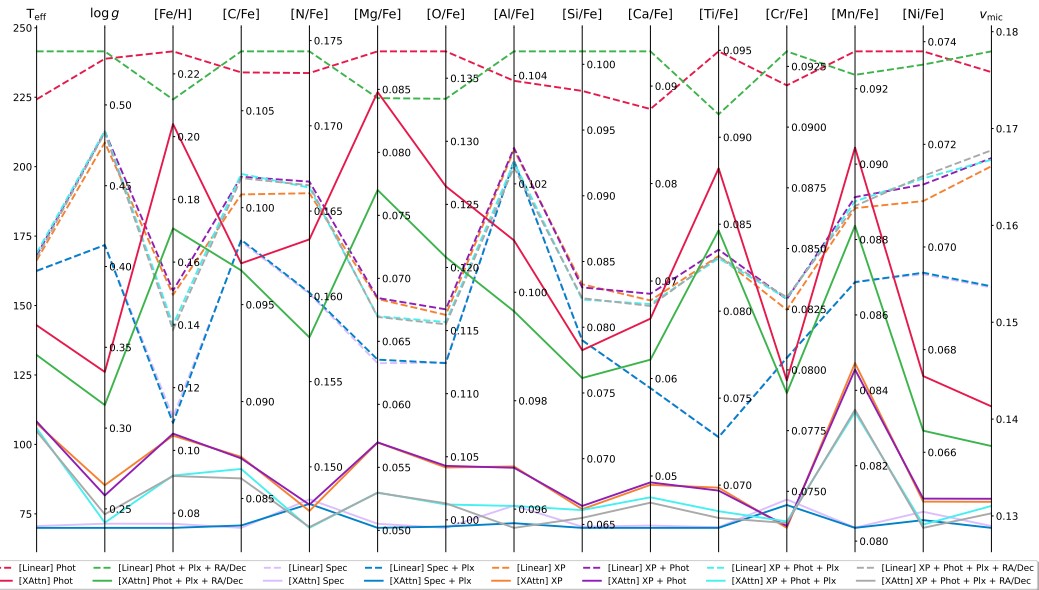

Figure 11: Predictive performance as measured by standard deviation of residuals on a held-out evaluation set of models with various input combinations on various physical properties of stars. **Lower is better.** Each line represents a different model, with dashed lines indicating mean-pooled linear probes and solid lines indicating cross-attention pooled linear probes. The color of the line represents the inputs that the model is given. All embeddings are generated from the frozen, pretrained encoder of Aion-1-L.

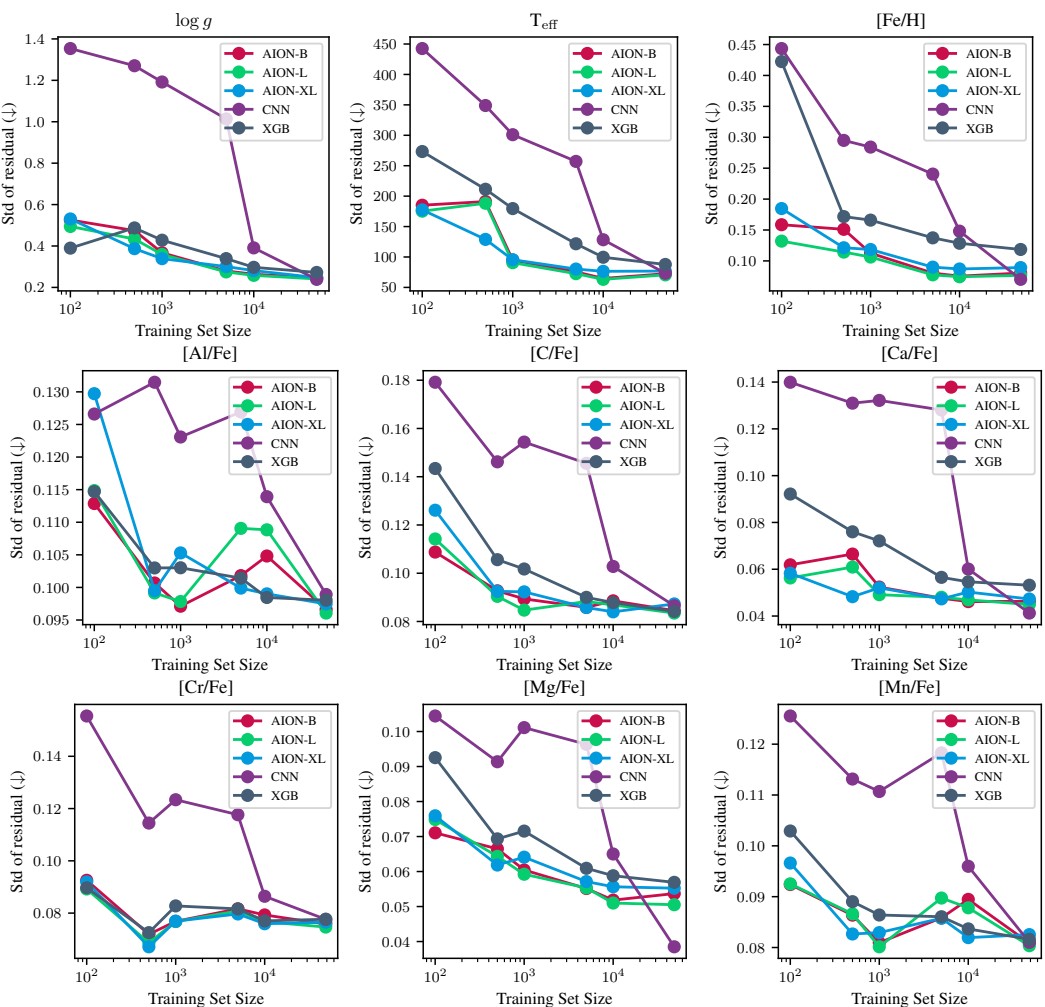

Figure 12: Predictive performance of AION-1-B/L/XL, a convolutional baseline from raw spectra, and an XGBoost baseline from spectrum tokens as a function of training set size for different stellar properties. Performance is measured in terms of the standard deviation of the residuals, and thus **lower is better**.

**Models.** For AION-1, we tokenize each cutout with the multi-survey image tokenizer, mean-pool the resulting embeddings[6], and pass the 768-d mean vector to a two-layer MLP head (hidden size = 256, GELU, dropout = 0.1). The head is trained on $80\%$ of the sample with class-stratified splits and evaluated on the remaining $20\%$.

**Baselines.** We replicate this protocol with the DINOv2 baseline, replacing the tokenizer with the ViT-g/14 backbone and applying the RGB normalization recommended by [47]. EfficientNet-B3 is trained end-to-end from random initialization using the same splits and standard data augmentations. Finally, we adapt ZooBot [69] by fine-tuning the penultimate layer on our $8,000$ samples; although ZooBot was never exposed to GZ10 labels, it benefits from pre-training on $\sim 300,000$ images covering the broader, harder GZ-5 decision tree, and thus acts as an approximate upper bound on achievable accuracy.

**Results.** As Table 4 shows, AION-1-L tops all baselines except ZooBot, exceeding EfficientNet by $+7.2$ pp and DINOv2 by $+15.8$ pp, while using only a lightweight MLP head. Moreover, it

---

[6]Although we experiment with attentive pooling in this setting, unlike with property estimation, we find that attentive pooling does not provide any meaningful gain in accuracy.

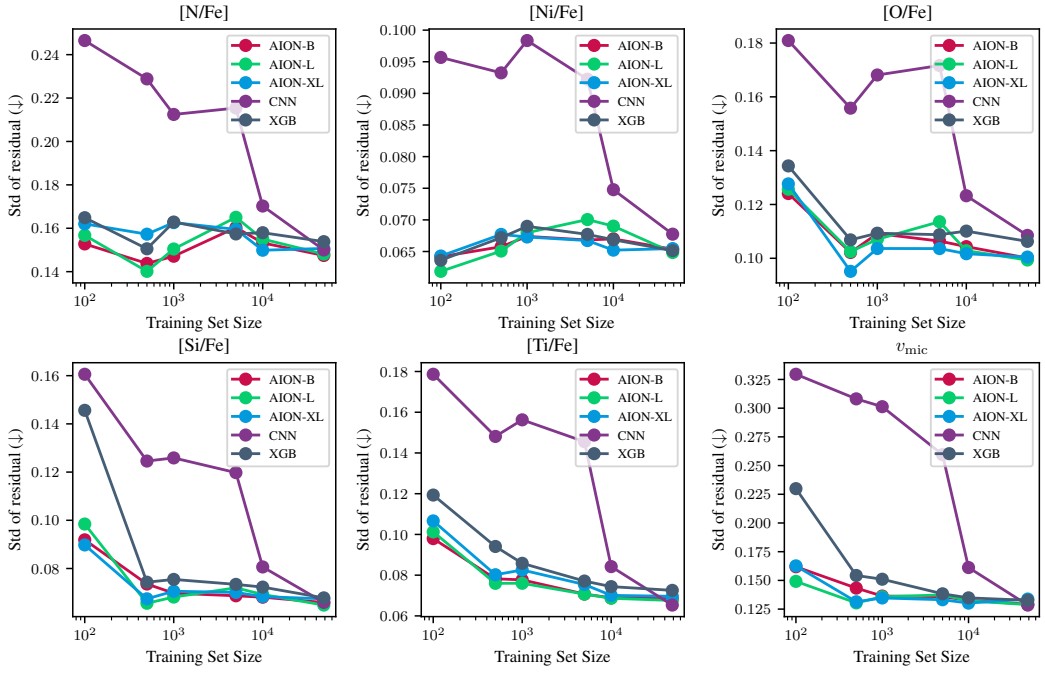

Figure 12: (continued)

reaches close to the ZooBot accuracy, only under-performing by $-2.4$ pp, despite seeing two orders of magnitude fewer labeled galaxy images during its inference process.

# G Details on Dense Predictions

In this section, we provide additional details on our dense fine-tuning experiments, which involve predicting segmentation maps and detecting sets of objects from image inputs. While prior work by [44, 7] classifies object detection as a sparse prediction task—treating it as an autoregressive sequence generation problem—we refer to it as dense prediction in contrast to our scalar prediction tasks. This distinction emphasizes the structured nature of segmentation and object detection compared to simpler regression-based outputs.

## G.1 Architecture

**Semantic Segmentation.** We implement a lightweight convolutional upsampler trained on top of AION-1's encoder representations. Our upsampler design is largely inspired by the mask decoder from [31][7], but with a key modification: we do not include hypernetworks instantiated from additional register tokens. Instead, we use a single convolutional layer to project the upsampled output to the desired number of segmentation maps, simplifying the architecture while maintaining efficiency.

**Clump Detection.** For clump detection, we introduce no additional model parameters. Instead, we finetune AION-1's decoder to autoregressively generate linearized object tokens, following the same tokenization scheme used in our pre-training catalog data. This approach enables the model to predict structured object sequences without requiring task-specific modifications.

## G.2 Galaxy Zoo 3D Segmentation

Galaxy Zoo 3D is a dataset derived from volunteer annotations of galaxies, originally presented in [39] and collected through the Zooniverse[8] citizen science platform. Each galaxy sample was

---

[7]https://github.com/facebookresearch/segment-anything/

[8]https://www.zooniverse.org/

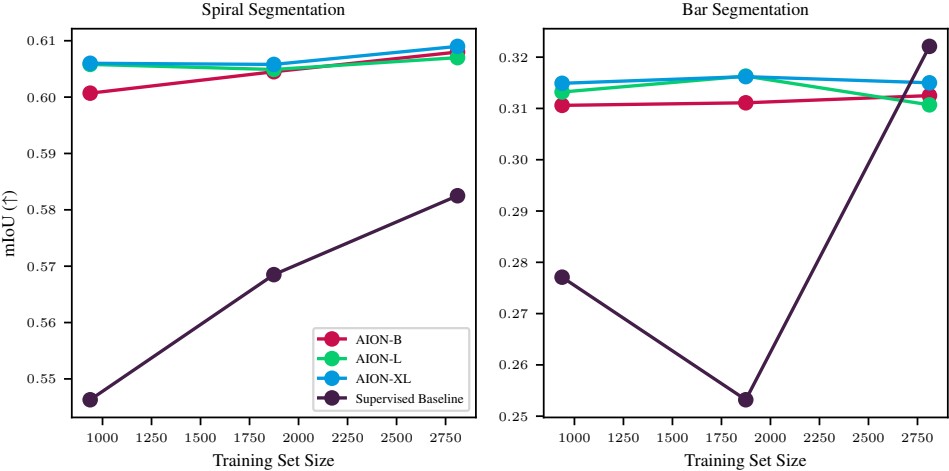

Figure 13: **IoU scores for spiral arm and bar segmentation** across different models, evaluated at three fractions of the available training data 33% (937), 66% (1874), and 100% (2811).

annotated by 15 volunteers, who were asked to mark the galactic center, any stars in the frame, and to draw bounding boxes around galactic bars and spiral arms.

For this study, we focus on the vote maps, which consist of four dense arrays containing pixel-wise annotation counts (ranging from 0 to 15) indicating how many annotators included a given pixel in their annotations. Segmenting bars and spiral arms is particularly challenging, which is why human annotations are crucial for this task.

Following [71], we filter out samples that do not reach a confidence level of 0.2 and compare our results to a model proposed in the same study, which we trained ourselves. The ground truth segmentation is defined as the set of pixels that received any number of votes from annotators, normalized by the maximum number of votes received per sample.

For our dense predictions, we finetune a small convolutional head on top of AION's frozen encoding module and optimize the trainable parameters using mean squared error (MSE).

We evaluate our model using the Intersection over Union (IoU) metric, reporting separate IoU scores for spiral arms and bars rather than a combined mean IoU (mIoU). The IoU is defined as:

$$\text{IoU} = \frac{|\hat{\mathbf{M}} \cap \mathbf{M}|}{|\hat{\mathbf{M}} \cup \mathbf{M}|}, \tag{9}$$

where $\hat{\mathbf{M}}, \mathbf{M} \in [0, 1]^{H \times W}$ denote the predicted and ground truth segmentation masks, respectively.

We determine separate segmentation thresholds for spiral arms and bars using 20% of our validation set. These thresholds are computed independently, as annotators tend to agree more consistently on bar structures than on spiral arms. This approach ensures optimal segmentation performance for both components.

### G.3 Galaxy Zoo Clump Detection

We additionally investigate the Seq2Seq problem of autoregressively generating galaxy clumps [17]. Specifically, we finetune AION-1 to generate an ordered sequence of clumps – where the number of clumps varies across different examples – by conditioning on Legacy Survey Images. To achieve this, we use the catalog tokenizer used during pre-training, which encodes catalog objects into a structured sequence of quintuples, each consisting of pixel coordinates, elliptical shapes, and radius. We then finetune the model with a causal language modeling objective, conditioning on both the previously generated clumps and the corresponding Legacy Survey images. This setup allows the model to learn spatial and morphological dependencies among clumps, ultimately improving its ability to generate realistic clump distributions for galaxy images. This dataset consists of 3727 cross-matched samples. Some qualitative examples are shown in Figure 14.

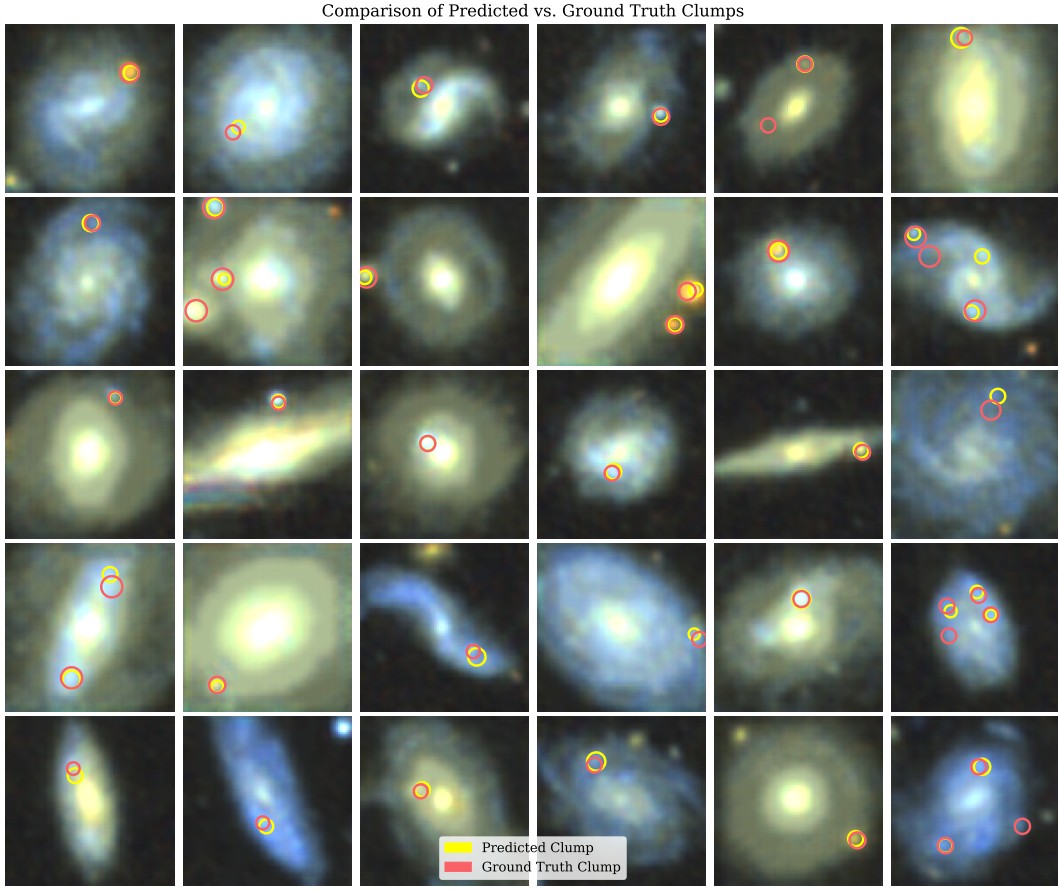

Figure 14: Qualitative examples of ground truth and predicted clump objects in Legacy Survey.

# H   Details on Retrieval Evaluation

## H.1   Evaluation Metric

For each query galaxy, we first produce an embedding using AION-1. We then generate embeddings for all other galaxies in the search corpus. All embeddings are averaged together to produce a single vector, $\mathbf{x} \in \mathbb{R}^d$, where $d$ is the embedding dimension of the specific AION model used. Next, we compute the cosine similarity between the query embedding, $\mathbf{x_q}$ and each candidate embedding $\mathbf{x_c}$, as

$$S_c(\mathbf{x_q}, \mathbf{x_c}) = \frac{(\mathbf{x_q})^T \mathbf{x_c}}{||\mathbf{x_q}||_2^2 ||\mathbf{x_k}||_2^2}. \tag{10}$$

The entire corpus (excluding the embedding) is then ranked in descending order of similarity. We compute the normalized Discounted Cumulative Gain (nDCG) retrieved objects, where the relevance, $r_i$ of each object, $i$, is determined by the criteria described in the sections below. Specifically, the DCG at rank $k$ is defined as:

$$\text{DCG@}k = \sum_{i=1}^{k} \frac{2^{r_i} - 1}{\log_2(i+1)}, \tag{11}$$

The ideal DCG (IDCG) is computed by sorting the items by descending relevance. The normalized DCG is then

$$\text{nDCG@}k = \frac{\text{DCG@}k}{\text{IDCG@}k}. \tag{12}$$

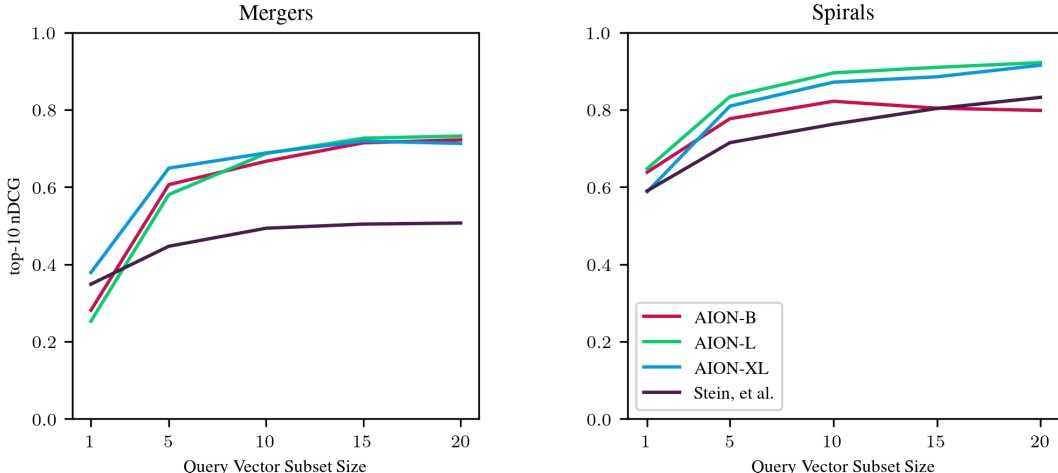

Figure 15: **nDCG@10 as a Function of Aggregated Query Size**. Overall, we find that mean-pooling multiple queries of the same object type to form our query vector dramatically improves model performance when searching for mergers and spirals, up to a query count of roughly 15. This holds for all models.

In our experiments, we focus on $k = 10$ and report $\text{nDCG@10}$ as our evaluation metric.

## H.2 Galaxy Zoo Object Labels

**Data.** We use the Galaxy Zoo DECaLS catalog [69], which provides citizen-science morphological classifications for a large set of galaxies observed in the Legacy Imaging Survey [16]. We then cross-match these classified objects with corresponding Legacy Survey South images. To ensure that each galaxy has sufficiently reliable volunteer annotations, we discard any objects with fewer than three total volunteer votes. This leaves $\approx 171,000$ galaxies.

Within this filtered dataset, we focus on two morphological classes of interest: *mergers* and *spirals*. We select high-confidence examples of these two classes by identifying galaxies for which more than $f = 0.9$ of the volunteers have voted for the corresponding morphology (merger or spiral). These high-confidence galaxies form our set of *query objects* in the retrieval experiments. In total, we have 726 merging galaxies and 24,622 spiral galaxies.

For each object in the dataset, we define a *relevance label* based on the fraction $f$ of volunteers who voted for the same morphological class. Thus, an object whose volunteer vote distribution aligns more strongly with merger or spiral morphology is assigned a higher relevance label than one whose distribution is more ambiguous. This setup encourages the retrieval model to prioritize both the correct morphological class and the degree of confidence in that classification.

**Aggregated Query Results.** In addition to evaluating single query objects, we investigate whether *aggregating* multiple queries can improve retrieval performance. Specifically, instead of using a single galaxy embedding as the query, we compute a single query embedding by *averaging* the embeddings of multiple galaxies from the same morphological class (merger or spiral). The rationale is that features indicative of the given morphology will be reinforced across several galaxy embeddings, while idiosyncratic features unrelated to that morphology will be muted.

Figure 15 shows how the nDCG@10 score varies with the number of query embeddings being averaged. We observe that performance systematically increases up to about 15 queries, after which gains plateau. In particular, for merger queries, this approach *more than doubles* the nDCG@10 compared to single-query retrieval, while for spirals it boosts nDCG@10 by approximately 0.25. Notably, even with aggregated queries, our best models still outperform the baseline method of [62], underscoring the effectiveness of this aggregation strategy.

|             | LS → LS | HSC → HSC | LS → HSC | HSC → LS |
| ----------- | ------- | --------- | -------- | -------- |
| AION-1-B    | 0.012   | 0.018     | 0.004    | 0.016    |
| AION-1-L    | 0.011   | **0.019** | 0.004    | **0.017** |
| AION-1-XL   | **0.015** | 0.015   | 0.004    | 0.012    |
| [62]        | 0.007   | –         | –        | –        |

Table 7: **nDCG@10 for Strong Lensing Retrieval.** We evaluate retrieval performance on strong gravitational lenses, measuring nDCG for galaxies retrieved via cosine similarity of AION-1's average token embeddings. Each column is labeled $\mathbf{X \to Y}$, $\mathbf{X}$ being the modality used to produce the query embedding and $\mathbf{Y}$ the key embedding. For all four columns, we show the results for the three AION-1 variants (B, L, XL). Since the state-of-the-art self-supervised baseline only generates legacy survey (LS) embeddings, we only show its results on the $\mathbf{LS \to LS}$ task.

## H.3  Strong Lens Finding

**Data.** For the strong lensing retrieval task, we start by filtering the cross-matched catalog of objects within the Legacy Survey and HSC datasets to approximately reproduce the parent sample used in the HSC strong lensing searches [29]. Specifically, we impose three additional cuts: (1) objects with photometric redshifts between 0.2 and 1.2, (2) objects with an estimated stellar mass above $5 \times 10^{10} M_\odot$, and (3) objects with a star formation rate to stellar mass ratio less than $1 \times 10^{-10}$. In order to identify the strong gravitational lenses within the resulting parent sample, we cross-match with previous lens-finding catalogs [57, 73, 58, 28, 59, 72, 29, 11, 55, 53, 35, 63, 8, 10, 9, 51, 45, 19, 26, 23, 46]. Most strong lensing catalogs offer a grade for each candidate. Since the criteria for this grading varies between catalogs, we ignore these grades and instead assign a relevance score of $1.0$ to all the strong lensing candidates found within each catalog. All other objects in our parent sample are given a relevance score of $0.0$. Even with the additional filtering, strong gravitational lenses make up only $0.1\%$ of our parent sample.

For each object in the parent sample, we have both the HSC and Legacy Survey observations. For AION-1, we extract the embeddings by passing the tokenized observation through the frozen encoder. We then average the output over the patch dimension. For the state-of-the-art baseline model [62], we directly use the representation output by the model. Unlike AION-1, the baseline is trained solely for legacy survey images. The resulting AION-1 embeddings have 768 dimensions for both modalities, whereas the baseline model embeddings have 128 dimensions for the Legacy Survey modality.

**Results.** Since we have both HSC and Legacy Survey (LS) embeddings for our strong lens catalog, we can perform two retrieval tasks within a modality: LS query with LS keys ($\mathbf{LS \to LS}$) and HSC query with HSC keys $\mathbf{HSC \to HSC}$. We can also explore two retrieval tasks between modalities: LS query with HSC keys ($\mathbf{LS \to HSC}$) and HSC query with LS keys ($\mathbf{HSC \to LS}$). The state-of-the-art baseline only enables $\mathbf{LS \to LS}$. The nDCG@10 metrics for these tasks are reported in Table 7.

We find that we outperform the state-of-the-art on the $\mathbf{LS \to LS}$ task for all three AION-1 model sizes, with the largest model leading to the greatest performance improvement. Switching from the $\mathbf{LS \to LS}$ task to the $\mathbf{HSC \to HSC}$ task leads to further gains in the retrieval metric, confirming that AION-1 is successfully leveraging the higher-resolution information present in the HSC images. For the cross-modality tasks we find mixed performance for AION-1. On the $\mathbf{LS \to HSC}$ task we get the worst retrieval performance of any modality combination, but on the $\mathbf{HSC \to LS}$ task we get equivalent retrieval performance to the $\mathbf{HSC \to HSC}$ task. One possible cause is that our retrieval depends on informative query embeddings. For example, in Appendix subsection H.2 we find that aggregating query embeddings leads to significant improvements in performance on galaxy morphology retrieval. For strong lensing retrieval, the HSC embeddings are derived from more informative (higher-resolution) observations than the LS embeddings. This leads to better performance when we query with the more informative HSC embeddings over the LS embeddings.

The low overall nDCG@10 score for all models and tasks reflects the inherent challenge in retrieving strong-lensing images. The Einstein ring that characterizes a strong gravitational lens is often dim compared to other features in the image, and strong lenses themselves are incredibly rare events. Despite this challenge, the state-of-the-art model we outperform has already been used to identify 1192 new strong lensing candidates [61].

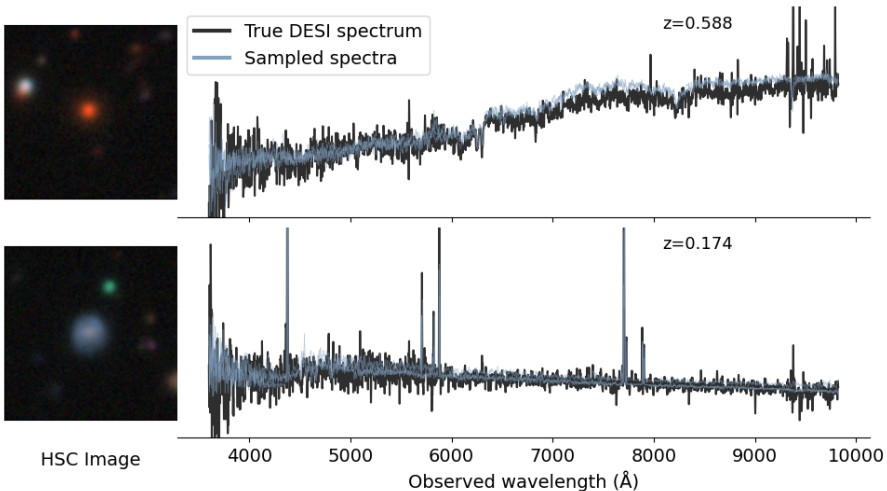

Figure 16: **Out-of-distribution conditional generation.** DESI spectra sampled from AION-1 (blue) conditioned on HSC images (insets), overlaid on the true DESI spectra (black). Even though HSC–DESI pairs were never seen during pre-training, the model reproduces key spectral features, demonstrating emergent transitive understanding.

## H.4    Emergent Transfer Properties

### H.4.1    Generative Transfer of Multimodal Understanding

Our training mixture contains many *pairwise* matches, e.g. HSC images $\leftrightarrow$ SDSS spectra and SDSS spectra $\leftrightarrow$ DESI spectra. However, the model is never trained to on the DESI-HSC pairs, and so never learns to produce HSC images from DESI spectra directly. Nevertheless, when we condition use AION-1 to sample a DESI spectrum from an HSC image, the generated spectrum (blue) closely tracks the ground-truth DESI spectrum (black); see Figure 16. Both a quiescent (red) galaxy and a star-forming (blue) galaxy are reproduced with realistic absorption and emission features, demonstrating that AION-1 has learned a transitive mapping across modalities. This is likely due to the fact that AION-1 already understands the mapping between HSC and other spectroscopic (SDSS) or imaging (Legacy Survey) surveys, as well as the mapping between those intermediate surveys and the final target survey, DESI.

### H.4.2    Survey-to-survey Transfer in Embedding Space

Beyond conditional generation, we ask whether AION-1's frozen image encoder produces survey-invariant representations that let us port knowledge from one telescope to another. Specifically, we train a single linear classifier on Legacy Survey embeddings to predict the ten Galaxy Zoo-10 morphology classes. The encoder weights remain fixed; only the 10-way soft-max layer is optimized. We then apply this exact head—*without any fine-tuning*—to embeddings of Hyper Suprime-Cam (HSC) images. To create the evaluation set we cross-match the HSC wide catalog (see subsection A.2 with Galaxy Zoo-10 (GZ10) volunteer votes and explicitly remove any targets that overlap with the Legacy Survey-GZ10 training set to prevent test leakage. The resulting sample contains $\sim$1,000 galaxies. As reported in Table 8, the zero-shot classifier attains $84 - 86\%$ accuracy across all AION-1 scales, essentially matching its performance on the native Legacy Survey domain. This robustness holds despite factor-of-$\sim$2 differences in depth, distinct filter sets ($\{g, r, i, z, y\}$ vs. $\{g, r, i, z\}$), and a different pixel scale. The result underscores that AION-1 embeddings capture morphology in a way that is largely agnostic to survey-specific imaging characteristics, enabling workflows that recycle scarce labeled data from one survey to bootstrap science in another.

|  | Legacy Survey (Train) | HSC (Eval.) |
|---|---|---|
| AION-1-B | 83.95 | 84.15 |
| AION-1-L | 87.16 | 85.66 |
| AION-1-XL | 86.99 | 85.91 |

Table 8: **Zero-shot morphology-classification accuracy (%).** A classifier trained on Legacy Survey images transfers directly to HSC, indicating that AION-1 produces survey-invariant representations.

## I  Details on Model Scaling

We evaluate how the model size impact its performances. The evaluation computes the categorical cross-entropy on predicted token outputs given input tokens that are randomly selected among every available modality. The overall evaluation loss is the weighted average of each modality loss. Figure 17 reports the overall evaluation loss and the image and spectrum losses for the legacy and SDSS surveys respectively. The decrease we see in the overall evaluation loss for the legacy survey indicates the model performs better when its size increases. However, the evaluation loss on SDSS remains similar regardless the size of the model. When checking for modality specifically (second row of Figure 17), it appears the image evaluation loss decreases with the size of the model while the spectrum evaluation loss stagnates. This indicates the amelioration observed while scaling the model is largely due to better performances in predicting token images. This trend could be explained by the fact that the complete dataset contains much more samples with images than with spectra (Figure 2a). There might not be enough spectrum data to observe improvement while scaling the model.

Additionally, we evaluate the impact of adding the GAIA dataset to the training set. GAIA contains 77M objects with spectrum data, thus should theoretically bring more information about this modality. Figure 17 compares the evaluation losses for different model sizes trained on all surveys except GAIA (Base Dataset i.e. Legacy survey + SDSS + DESI + HSW) and on all surveys including GAIA (Base Dataset + GAIA). For both evaluations on Legacy and SDSS surveys the loss is higher when training with Base Dataset + GAIA. Adding GAIA survey to the training dataset seems thus to impacts negatively the performances of the model on the other surveys. It is the case for image and even spectrum modality. The decreased performance on spectrum modality of SDSS survey, while adding spectrum information from GAIA in the training dataset, could be explained by the fact that GAIA spectrum is of much lower resolution than the one of SDSS.

## J  Full Modality Tokens

| Category | Description | Modality |
|---|---|---|
| Imaging (2) | Legacy Survey imaging | tok_image |
| | HSC Wide imaging | tok_image_hsc |
| Catalog (1) | Legacy Survey catalog | catalog |
| Spectra (2) | SDSS spectra | tok_spectrum_sdss |
| | DESI spectra | tok_spectrum_desi |
| Gaia (4) | Gaia BP spectrum | tok_xp_bp |
| | Gaia RP spectrum | tok_xp_rp |
| | Gaia parallax | tok_parallax |
| | Sky coordinates | tok_ra, tok_dec |
| Gaia Photometry (3) | Gaia G-band flux | tok_flux_g_gaia |
| | Gaia BP-band flux | tok_flux_bp_gaia |
| | Gaia RP-band flux | tok_flux_rp_gaia |
| Legacy Survey (9) | g-band flux | tok_flux_g |
| | r-band flux | tok_flux_r |
| | i-band flux | tok_flux_i |
| | z-band flux | tok_flux_z |
| | WISE W1 flux | tok_flux_w1 |
| | WISE W2 flux | tok_flux_w2 |
| | WISE W3 flux | tok_flux_w3 |
| | WISE W4 flux | tok_flux_w4 |
| | E(B-V) extinction | tok_ebv |
| Legacy Survey Shape (3) | Ellipticity component 1 | tok_shape_e1 |
| | Ellipticity component 2 | tok_shape_e2 |
| | Effective radius | tok_shape_r |
| HSC Photometry (5) | g-band magnitude | tok_mag_g |
| | r-band magnitude | tok_mag_r |
| | i-band magnitude | tok_mag_i |
| | z-band magnitude | tok_mag_z |
| | y-band magnitude | tok_mag_y |
| HSC Extinction (5) | g-band extinction | tok_a_g |
| | r-band extinction | tok_a_r |
| | i-band extinction | tok_a_i |
| | z-band extinction | tok_a_z |
| | y-band extinction | tok_a_y |
| HSC Shape (3) | Shape component 11 | tok_shape11 |
| | Shape component 22 | tok_shape22 |
| | Shape component 12 | tok_shape12 |
| Other (1) | Redshift | tok_z |

Table 9: Complete list of modalities used in our multi-survey analysis. The modalities are grouped by their source surveys and measurement types. Numbers in parentheses indicate the count of modalities in each category.

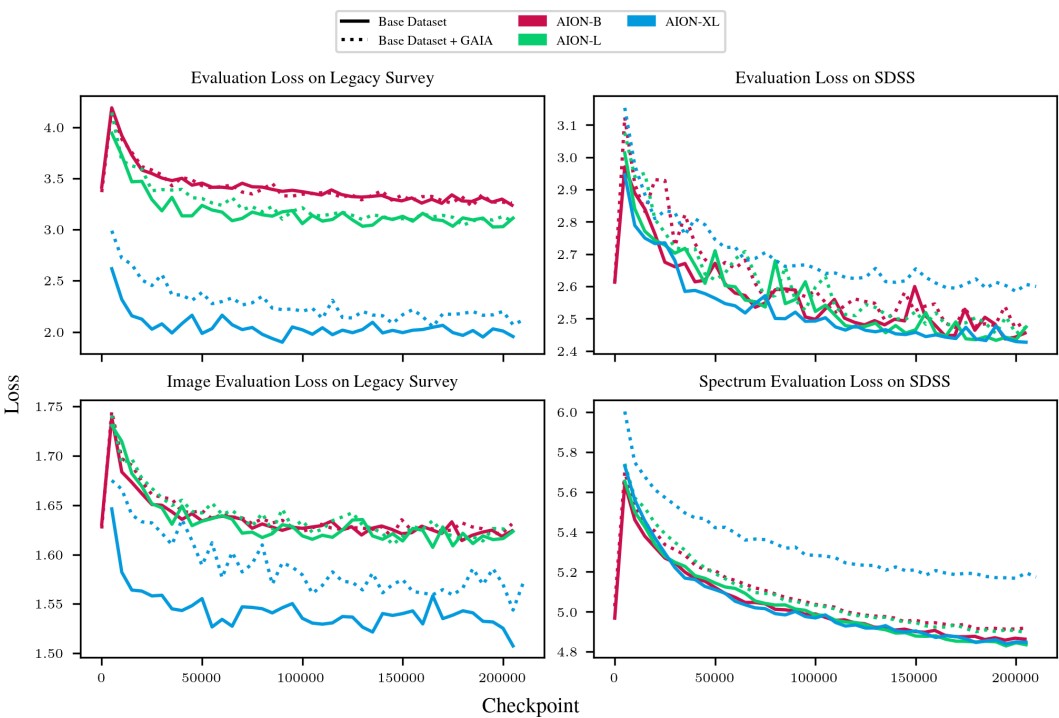

Figure 17: Evaluation losses for different model sizes on Legacy and SDSS surveys including image and spectrum modality. Base dataset refers to the training on all surveys except GAIA, while Base Dataset + GAIA are the models trained on all surveys described in Figure 2a.

