# OpenReview forum: "AION-1: Omnimodal Foundation Model for Astronomical Sciences"
_NeurIPS.cc/2025/Conference — NeurIPS 2025 poster_

### Official Review · Reviewer_KFN2 · 2025-07-02

**Clarity:** 3
**Significance:** 3
**Originality:** 2
**Rating:** 5
**Confidence:** 5

**Summary:**

The paper describes the implementation of a multimodal model for astronomical sources, trained on curated astronomy data. It introduces a specialized tokenization scheme to train multimodal masked modeling, and shows multiple astronomy analysis tasks to assess the quality of the trained model.

**Questions:**

1. Would the authors consider adding limitations (not just selection biases) in the main text, perhaps with an hallucinated example?
2. I understand ablation studies justifying choices can be very costly for these type of models, but would the authors add text justifying the choices of tokenizer and models, and perhaps give a short list of the things which did not improve or did not work?
3.  Decrease of performance when adding Gaia spectra to the other modalities: authors explanation blame the lower resolution, even though it add information. This would perhaps require a bit more explanation or proof that is not an actual limitation of the chosen architecture.

**Ethical Concerns:**

["NO or VERY MINOR ethics concerns only"]

**Final Justification:**

I increased my score. I was pretty close to a higher score initially and the authors addressed enough (limited metrics, missing limitations) to raise it.

**Limitations:**

Yes.

**Paper Formatting Concerns:**

Not really a formatting issue, but appendix is long.

**Quality:**

3

**Strengths And Weaknesses:**

**Strengths**
* Performant model evaluated on a large set of astronomy analysis tasks
* Large engineering effort, and well detailed (in appendix)
* Tokenization framework is interesting

**Weaknesses**
* Could show the limits of the model: where it hallucinates, where it underperforms
* Most results are in the appendix. Actually the appendix is quite a long read, which makes me think if this is the right venue.
* No ablation study at either modelling or tokenization is shown
* Evaluation metrics could be better chosen. $R^2$ is not well suited (perhaps adjusted-$R^2$ and showing a dispersion metric would help)
* Some overselling that does not help the text or is not convincing

*Various small issues*:
- line 57: "AION-1 is the first large-scale model designed for arbitrary combinations of highly heterogeneous scientific observations"  no it is not. Weather, remote sensing, biology models have already been developed and published with multi-modal data
- line93:  "two key challenges: the variety of data types (2D images, 1D spectra, scalar values) and the diversity of sources within each type (different telescopes, resolutions, and instrument formats)." I do not see how variety and source diversity make it special. Natural images also come from different camera, formats, resolutions...I think the authors mean the quantitative tasks they are doing with the data are indeed much different and could require more careful on the source, but data diversity or variety is not really more heterogeneous for any other domain
- "Built on a ResNet backbone adapted from [71]" : reference 71 points to MagViT which does not use ResNet but VQGAN
 line237: "While we see that AION-1 performs competively against supervised baselines in the full-data regime, its performance is much stronger in the low-data regimes, where it matches or surpasses traditional models that require an order of magnitude more training data" : the authors meant "low-label regimes" and "order of magnitude more labeled data"
- Figure 2a: no links between Legacy survey and HSC or between Legacy Survey and Gaia: was this an omission?

---

> ### Author Rebuttal · Authors · 2025-07-29
>
> We would like to thank the reviewer for the detailed review and insightful feedback.
>
> ## Model Limitations/Hallucinations
>  Thank you for highlighting the need to characterize where the model under‑performs. We will add some examples discussing hallucination to the paper. For example, when we ask AION‑1 to super‑resolve low‑resolution LegacySurvey images into high‑resolution HSC imaging, the model occasionally “hallucinates” faint background galaxies that are not present in the ground truth. This behavior is an illustration that the conditional distributions between complex modalities are not perfectly captured, which is always to be expected (no machine learning model is perfect). We are aware of these imperfections in a generative framework, as well as the need for using AION-1 in a framework enforcing explicit conditioning on observed data with an explicit likelihood term which prevents hallucinations. We will clarify this in the main body of the text, and emphasize that in the absence of this framework we envision using AION-1 primarily an embedding model, which will be calibrated for downstream tasks with linear/attention heads that are trained on the specific task; this strategy to use the model is not subject to hallucinations.
>
> ## Quality of Evaluation Metrics
> Using R^2 is the primary regression metric in recent multimodal‐astronomy studies such as AstroCLIP (Parker, et al., 2024) and MAVeN (Zhang, et al., 2024). However, we agree with the reviewer that a dispersion‑based measure is also valuable. Accordingly, we have re‑evaluated galaxy‑ and stellar‑property predictions using the standard deviation of the residuals (see tables below). We will also add residual‑dispersion plots in the appendix for visual inspection.
>
> Galaxy Measurements -> Property (Std. Dev. of Residuals):
> | Model | $z$ | $M_\text{star}$ | $t_\text{age}$ | $Z_\text{met}$ | $\text{SFR}$  |
> |---|---|---|---|---|--|
> | AION-1-B (Ph) | 0.056 | 0.354 | 1.326 | 0.694 | 1.778 |
> | AION-1-B (Ph+Im) | 0.029 | 0.228 | 1.226 | 0.648 | 1.360 |
> | AION-1-B (Ph+Im+sp) | 0.008 | 0.141 | 1.128 | 0.566 | 1.195 |
>
> Stellar Spectra -> Property (Std. Dev. of Residuals):
> | Model | $T_{eff}$ | $\text{logg}$ | $v_\text{mic}$ | $\text{Fe/H}$  |
> |---|---|---|---|--|
> | AION-1-B | 72.48 | 0.246 | 0.129 | 0.080 |
> | Baseline | 73.28 | 0.239 | 0.129 | 0.070 |
>
>
> ## Ablation Studies
> For the 4M architecture and training choices, we rely on the extensive ablation studies reported in 4M and therefore adopt the same encoder-decoder transformer configuration without re‑running those costly sweeps. Our own ablations focus on the new tokenisers. In the appendix we present a codebook‑size sweep for the image tokenizer: sizes ranging between 2^4 and 2^14 show a steady increase in performance up to 2^12, after which gains in performance begin to saturate. We therefore selected 2^	12 as the size for the image tokenizer, and fixed the other tokenizer sizes according to this based on a rough estimation of the relative complexity of that data.
>
> ## Other Issues
> **Length of Appendix:** We followed the NeurIPS 9‑page limit by keeping the core findings in the main manuscript, while placing extensions in the appendix for completeness. For example, galaxy‑property prediction with cross‑attention pooling appears in the main text (Table 2); the appendix then adds a comparison to the mean‑pooling variant. A similar approach is used when presenting the rest of the results.
>
> **Overselling **: Thank you for flagging wording that felt overstated. In the camera‑ready version we will remove superlatives that are not strictly supported by evidence and address the individual points raised below.
>
> **Line 57: “AION-1 is the first [...]”:** We will delete the sentence at line 57 and instead clarify that AION‑1 is, to our knowledge, the first omnimodal foundation model for astronomy - while multimodal models already exist in weather, remote sensing, and biology.
>
> **Line 93: “two key challenges [...]”:** We agree that natural‑image corpora also features multiple cameras, formats and resolutions, so the phrasing was imprecise. What is distinctive in astronomy is the combination of cross‑dimensional heterogeneity (2‑D images, 1‑D spectra, irregular time‑series, scalar catalogue values) with instrument‑specific systematics that vary by orders of magnitude across wavelength, aperture and detector type. These differences carry physical meaning (flux calibration, spectral resolution, cadence) and directly affect the quantitative tasks we target (e.g. redshift, stellar‑mass inference). We will revise the sentence to emphasise this physical and dimensional heterogeneity rather than implying that source diversity alone makes astronomy unique.
>
> **“Built on a ResNet backbone [...]”:** We thank the reviewer for pointing this out. VQGAN does have residual blocks but it’s not technically built on a ResNet backbone; we will modify the wording in the final paper to reference VQGAN instead.
>
> **Line 237: “While we see that AION-1 performs [...]”:** Thank you for pointing this out. We will replace “low‑data regimes” and “training data” with “low‑label regimes” and “labelled data” to accurately reflect that the advantage is in settings with scarce annotations, not limited raw observations.
>
> **Figure 2a: no links [...]:** No link is drawn between our Legacy Survey catalog and our Gaia catalog because the LS catalog specifically targets galaxies, while the Gaia catalog targets stars; consequently, no individual objects are explicitly targeted by both catalogs.

---

> > ### Comment · Area_Chair_1U7t · 2025-08-05
> >
> > Hi,
> >
> > As the deadline of the discussion is approaching, could you please check the authors' rebuttal and respond accordingly?
> >
> > Thanks
> >
> > AC

---

### Official Review · Reviewer_1y7Y · 2025-07-02

**Clarity:** 3
**Significance:** 3
**Originality:** 2
**Rating:** 5
**Confidence:** 4

**Summary:**

This paper introduces AION-1, which is a family of large multimodal foundation models designed for astronomy’s diverse data landscape. Unlike prior approaches that often leverage natural language to connect modalities, AION-1 directly integrates up to 39 heterogeneous data modalities within a single unified model. Interestingly, after unsupervised pretraining, AION-1 demonstrates several emergent capabilities without task-specific finetuning. For example, its learned latent space encodes rich astrophysical information that can be exploited by simple linear probes: on predicting galaxy properties (redshift, stellar mass, etc.) and stellar parameters, frozen AION-1 embeddings achieve performance on par with or exceeding dedicated supervised models, especially when only limited labeled examples are available. Notably, in low-data regimes (tens to a few hundred samples), AION-1’s representations often surpass specialized models that were trained from scratch on each modality.

**Questions:**

AION-1 is purposely made for arbitrary combinations of data modality, but how does it handle objects with substantially longitudinal missing data? For example, if an object has only photometry and no images or spectra, how does the model perform?
In a similar vein, beyond the spectral super-resolution example, can AION-1 perform more exotic cross-modal generation?
What was the motivation against fine-tuning the entire AION-1 model on a downstream task?  Is it due to a loss in generality?

**Ethical Concerns:**

["NO or VERY MINOR ethics concerns only"]

**Limitations:**

one of the biggest weaknesses/limitations of the paper is the lack of accounting of temporal dynamics. This is indeed crucial for certain domains (e.g. monitoring variable stars or exoplanet transits). A suggestion is to develop a time-series tokenizer that could, for example, convert a light curve into a set of tokens (perhaps via discrete wavelet coefficients or sequence VQ encoding). The authors do acknowledge this fervently, and I do hope to see future work on this.

A weakness not touched upon is interpretability. Foundation models can be black boxes, however, in science it’s important to extract understanding. The paper does a bit of this by showing latent space structure and that certain attention heads attend to spectral lines (in SpectraFM style), but I believe more could be done. Maybe through a more principled approach towards taking in the data, or by providing tools or visualizations for scientists to interpret what AION-1 has learned (e.g. which token dimensions correspond to physical properties). This would increase trust and adoption in the community.

**Paper Formatting Concerns:**

generally well-written and organized

**Quality:**

3

**Strengths And Weaknesses:**

Strengths:
The paper’s methodology is sound and represents a rigorous application of state-of-the-art self-supervised learning to a novel domain. The authors carefully address the challenges of heterogeneous data: each modality is standardized via appropriate tokenization (e.g. ResNet+VQ for images, ConvNeXt for spectra) with built-in awareness of noise and units.
Their models are comprehensively tested and across the board show very strong performance against other standard models, particularly shining when data is scarce (as is often the case in astronomy).
AION-1 is also a significant step up at the intersection of AI and astronomy. It is the first model to unify such an expansive range of data modalities in astronomy and scale to the billion-parameter regime. This is a substantial step from earlier efforts like AstroCLIP or PAPERCLIP.

Weaknesses:
While the experiments are extensive, one area that could strengthen the quality further is full fine-tuning or end-to-end evaluation on downstream tasks. The study primarily uses frozen representations with simple heads (or small added conv layers) to probe performance.
Another minor concern is that the model’s treatment of temporal data remains absent but the authors acknowledge this as a current limitation.
The novelty in algorithmic terms is somewhat limited. The architecture is an encoder-decoder Transformer with a BERT-like objective. Maybe more specific physics informed elements could have been incorporated, but overall the methods are motivated well.

---

> ### Author Rebuttal · Authors · 2025-07-29
>
> We sincerely thank the reviewer for the thoughtful, detailed feedback and for recommending acceptance. We are encouraged that you found the methodology rigorous, the omnimodal tokenization sound, and the low‑data performance particularly impactful.
>
> ## Handling Heavily Missing Modalities
> AION‑1’s masked‑token pre‑training objective is modality‑agnostic: at both training and inference, any modality can be absent and the sequence simply contains no tokens of that type. Empirically, this robustness holds: when we feed photometry only for the galaxy‑property benchmark, AION‑1 still outperforms an XGBoost model trained on the same photometric features (Table 2). We ran the same ablation for stellar‑parameter prediction, again supplying only photometry, and similarly found that AION-1 generally outperforms an XGBoost baseline.
>
> | Model | $T_\text{eff}$   | $\log g$ | $[Fe/H]$ |
> |---|---|---|---|
> | AION-L + Attn  | **0.946**| **0.955** | 0.582 |
> | XGBoost | 0.943  | 0.950 | **0.589**  |
>
> ## More exotic cross-modal generation
> Thank you for raising this point, and we agree that it would be interesting to see if we can perform more exotic cross-modal generations. As a meaningful test, we considered a combination of modalities that are never seen during pre-training: HSC imaging and DESI spectra. Our training strategy included pairs of modality examples HSC imaging <-> SDSS spectra and SDSS spectra <-> DESI spectra, but never directly seeing HSC imaging <-> DESI spectra. This allows us to test whether the model is building a transitive understanding of data modalities which would ultimately allow us not to have a combinatorial explosion of modality combinations needed for training.
>
> At test time, asking the model to generate DESI spectra from HSC imaging (which, again, is an out of distribution task) produces physically plausible DESI‑resolution spectra whose key absorption/emission features align with the ground truth for both quiescent red galaxy and a star forming blue galaxy. We will include examples of these generations in the revision of the paper.
>
> ## Temporal Data
> We thank the reviewer for emphasizing the need to incorporate time‑domain information and fully agree that incorporating this data in the future will be critical, allowing us to extend our model to a range of use cases from stellar variability to supernovae characterisation. As a matter of fact, our AION-2 model, currently under development, includes a dedicated time-series tokenizer.
>
> ## Finetuning
> We explore using AION‑1 primarily as a frozen embedding resource as we try to match the low-data, low‑compute workflow we expect most astronomers to adopt. Indeed, the main point we want to make to the community is that adapting such Foundation Model to their use case is trivial. And  we  note that the cross-attention heads on the frozen embeddings already surpass supervised baselines on most of the tasks that we evaluate on. Nonetheless, the transformer can be fine‑tuned end‑to‑end, and we are including utilities in our open source library to enable several fine-tuning strategies (from only adapting embedding layers to full fine-tuning) which we expect will be useful to adapt the model to  tasks and modalities that depart substantially from those we evaluated.
>
> ## Interpretability
> We agree this is an important and exciting future direction. One avenue we are already exploring is training sparse autoencoders on the latent space to disentangle which dimensions capture specific physical features, and we look forward to reporting on these results in future work. More generally, we recognise that neural networks in science must be applied with methodological care, so our workflow always includes a dedicated calibration stage: a small, carefully curated downstream‑task training set is used to train a calibration head on the embeddings for the specific physical question, ensuring that the learned representations remain trustworthy.

---

> ### Comment · Area_Chair_1U7t · 2025-08-05
>
> Hi,
>
> As the deadline of the discussion is approaching, could you please check the authors' rebuttal and respond accordingly?
>
> Thanks
>
> AC

---

### Official Review · Reviewer_SExS · 2025-07-03

**Clarity:** 3
**Significance:** 3
**Originality:** 1
**Rating:** 3
**Confidence:** 1

**Summary:**

This paper presents AION-1, the first large-scale, multimodal foundation model for astronomy. The authors propose a two-stage architecture to handle the diverse data types in this domain, such as images, spectra, and tabular data. First, modality-specific tokenizers convert the heterogeneous inputs into a common discrete token space. Then, a single transformer-based model is trained using a masked modeling objective on sequences of these tokens from over 200 million astronomical objects. The main contribution is a unified framework that can ingest arbitrary combinations of astronomical data to perform a wide range of downstream tasks, including parameter regression, classification, data generation, and object retrieval.

**Questions:**

The introduction makes a strong claim about the superiority of masked modeling over contrastive learning for this scientific domain. However, the experimental comparisons are primarily against supervised, single-modality baselines. To truly substantiate this claim, a more direct comparison would be beneficial. Could you discuss how AION-1's representations or downstream performance compare to a strong contrastive model, like a CLIP-style architecture, trained on the same multimodal data (e.g., images and spectra)?

**Ethical Concerns:**

["NO or VERY MINOR ethics concerns only"]

**Limitations:**

yes

**Quality:**

3

**Strengths And Weaknesses:**

I am not familiar with the area and don't know how to judge such a paper.

Pros
* It’s especially impressive in situations where you don't have a lot of data, which is a common headache in scientific research.
* Emergent Abilities: It can do some really cool things that aren't just simple predictions. For example, it can create high-resolution spectra from low-res ones and is great at finding rare objects, which really showcases its advanced capabilities.
* Clear and Well-Written: the paper easy to follow.
* Open and Accessible: they are open-sourcing everything – the models, the data, and the code.

Cons:

* **Not Entirely New:** The core idea isn't brand new. It builds heavily on recent models like the 4M model. The main innovation here is applying these ideas to a new scientific field and the clever engineering they did, not a completely new machine learning concept.
* **Insanely Expensive to Train:** The amount of computer power needed to train this model is a major practical drawback. To their credit, the authors admit this and suggest that future work could focus on making it more efficient.

---

> ### Author Rebuttal · Authors · 2025-07-29
>
> We would like to thank the reviewer for the detailed review and insightful feedback.
>
> ## 1. Methodological Novelty
> The NeurIPS 2025 Call for Papers explicitly lists “Machine learning for sciences” among its core topical areas, so contributions that unlock new scientific capabilities with ML fall squarely within scope. Although AION‑1 adopts the two‑stage tokenizer‑plus‑transformer design introduced by 4M, it delivers a decisive advance for astronomy. Earlier astronomical FMs, such as AstroCLIP (galaxies; Parker, et al., 2024) and Maven (supernovae; Zhang, et al., 2024), use contrastive losses to align only 2-3 modalities, must train on the small subset of objects where those modalities co‑exist, and target a single physical process. AION‑1, by contrast, scales to 39 modalities, enables training over all available data, and is the first model in astronomy to reach the billion-parameter scale and to process multiple physical phenomena simultaneously (e.g. galaxies, stars, and quasars). This omnimodal approach tackles astronomy’s inherently heterogeneous, weakly overlapping datasets, making AION‑1 the first foundation model capable of scaling across the entire domain, and it already sets new state‑of‑the‑art marks on a variety of tasks.
>
> ## 2. Cost of Training
> The reviewer is correct that pre‑training AION‑1 is non‑trivially expensive. However, its compute footprint is well inside the norms already accepted for scientific foundation models. Our largest release, AION-1-XL, is trained for 3.5 days on 288 H100 GPUs, equivalent to roughly 1000 GPU-days; by comparison, the ESM-2 3B protein model used 512 V100s for ~30 days (15,000 GPU-days), and the 15B version doubled this budget.
>
> Moreover, while the compute cost is high, it is a one-off expense that will be amortized by community use, especially relative to the entire zoo of single-use deep learning models trained for end-to-end specific tasks throughout astronomy. We will release the full pretrained weights of the model, tokenizers, and all model code, enabling users to easily use the model embeddings for downstream tasks, which we demonstrate already beat end‑to‑end supervised baselines on a variety of benchmark tasks.
>
> In short, the upfront compute is comparable to other successful scientific FMs, and the community can inherit the benefits at a tiny fraction of the cost.
>
> ## Comparison with CLIP-style architecture
> To address the reviewer’s request, we benchmarked AION‑1 against the strongest publicly‑available contrastive model for astronomy, AstroCLIP (Parker, et al., 2024), and an out‑of‑the‑box computer‑vision baseline, DINOv2‑B (Oquab, et al., 2023). All models were evaluated on the same train/test splits and galaxy‑property tasks.
>
> Galaxy Images -> Property (R^2)
> | Model | $z$ | $M_\text{star}$ | $t_\text{age}$ | $Z_\text{met}$ | $\text{SFR}$  |
> |---|---|---|---|---|--|
> | AION-B + Attn | **0.934**  | **0.886** | **0.445** | **0.490** | **0.637** |
> | AstroCLIP | 0.783  | 0.725 | 0.293 | 0.362 | 0.424 |
> | DINOv2-B | 0.571  | 0.550 | 0.166 | 0.284 | 0.251|
>
> Galaxy Spectrum -> Property (R^2)
> | Model   | $z$   | $M_\text{star}$ | $t_\text{age}$ | $Z_\text{met}$ | $\text{SFR}$  |
> |---|---|---|---|---|---|
> | AION-B + Attn | **0.995** | **0.956** | **0.532** | **0.610**  | **0.720** |
> | AstroCLIP | 0.985  | 0.898 | 0.518 | 0.601  | 0.704 |
>
> Galaxy Image Rare Object Retrieval (nDCG@10)
> | Model | Spirals | Mergers | Strong Lenses |
> |---|---|---|---|
> | AION-XL | **0.621** | **0.384**  | **0.015**  |
> | AstroCLIP | 0.602   | 0.248   | 0.006  |
> | DINOv2-B  | 0.477   | 0.060   | 0.003  |
>
> Ultimately, we find that AION-1’s embeddings outperform those produced by both AstroCLIP and DINOv2 in terms of R^2 for the property prediction tasks and nDCG@10 for the retrieval tasks.

---

> > ### Comment · Reviewer_SExS · 2025-08-05
> >
> > I thank the authors for their rebuttal. I think my concern about novelty still exists. However, I am not familiar with the area and I am not sure whether this application is enough or qualified. Thus I decide to keep my score. The AC can take this into consideration when looking at my rating. Thanks.

---

> ### Comment · Area_Chair_1U7t · 2025-08-05
>
> Hi,
> As the deadline of the discussion is approaching, could you please check the authors' rebuttal and respond accordingly?
>
> Thanks
> AC

---

### Decision · Program_Chairs · 2025-09-17

**Decision:**

Accept (poster)

**Comment:**

The paper presents a family of multimodal foundation models for astronomy, named AINO-1.
The final ratings are two "accept" and one "borderline reject".
Reviewer KFN2 raised the score to “accept” after the author's rebuttal addressed the concerns of limited metrics and missing limitations.
Reviewer SExS maintained the score of "borderline reject" considering the limited novelty, while also noting that the reviewer is not familiar with the area.
The AC acknowledges that this work introduces a unified framework and baselines into the area of astronomy. The results looks good.
Thus, AC would like to suggest to accept.
However, the authors need to revise the paper according to all reviews.